# Genetic architecture of natural variation of cardiac performance from flies to humans

**Saswati Saha**[1†], **Lionel Spinelli**[1†], **Jaime A Castro Mondragon**[2], **Anaïs Kervadec**[3], **Michaela Lynott**[3], **Laurent Kremmer**[1‡], **Laurence Roder**[1], **Sallouha Krifa**[1], **Magali Torres**[1], **Christine Brun**[1,4], **Georg Vogler**[3], **Rolf Bodmer**[3], **Alexandre R Colas**[3]*, **Karen Ocorr**[3]*, **Laurent Perrin**[1,4]*

[1]Aix-Marseille University, INSERM, TAGC, Turing Center for Living systems, Marseille, France; [2]Centre for Molecular Medicine Norway (NCMM), Oslo, Norway; [3]Development, Aging and Regeneration Program, Sanford Burnham Prebys Medical Discovery Institute, La Jolla, United States; [4]CNRS, Marseille, France

**Abstract** Deciphering the genetic architecture of human cardiac disorders is of fundamental importance but their underlying complexity is a major hurdle. We investigated the natural variation of cardiac performance in the sequenced inbred lines of the *Drosophila* Genetic Reference Panel (DGRP). Genome-wide associations studies (GWAS) identified genetic networks associated with natural variation of cardiac traits which were used to gain insights as to the molecular and cellular processes affected. Non-coding variants that we identified were used to map potential regulatory non-coding regions, which in turn were employed to predict transcription factors (TFs) binding sites. Cognate TFs, many of which themselves bear polymorphisms associated with variations of cardiac performance, were also validated by heart-specific knockdown. Additionally, we showed that the natural variations associated with variability in cardiac performance affect a set of genes overlapping those associated with average traits but through different variants in the same genes. Furthermore, we showed that phenotypic variability was also associated with natural variation of gene regulatory networks. More importantly, we documented correlations between genes associated with cardiac phenotypes in both flies and humans, which supports a conserved genetic architecture regulating adult cardiac function from arthropods to mammals. Specifically, roles for PAX9 and EGR2 in the regulation of the cardiac rhythm were established in both models, illustrating that the characteristics of natural variations in cardiac function identified in *Drosophila* can accelerate discovery in humans.

## Editor's evaluation

The authors investigated natural variation and new genetic mechanisms underlying cardiac performance using sequenced inbred lines of the *Drosophila* Genetic Reference Panel. The study provides insights into the genetic architecture of complex cardiac performance traits and represents an important resource for researchers studying cardiac performance.

## Introduction

Heart diseases is a major cause of mortality (*Bezzina et al., 2015*). Although a large number of genome-wide association studies (GWAS) have identified hundreds of genetic variants related to cardiovascular traits (*Roselli et al., 2018*; *van Setten et al., 2018*; *Shah et al., 2020*; *Verweij et al., 2020*), we are very far from a comprehensive understanding of the genetic architecture of these

*For correspondence:
acolas@sbpdiscovery.org (ARC);
kocorr@SBPdiscovery.org (KO);
laurent.perrin@univ-amu.fr (LP)

†These authors contributed equally to this work

Present address: ‡INRAE, Institut Sophia Agrobiotech, Université Côte d'Azur, CNRS, Sophia Antipolis, France

Competing interest: The authors declare that no competing interests exist.

complex traits. Deciphering the impact of genetic variations on quantitative traits is however critical for the prediction of disease risk. But disentangling the relative genetic and environmental contributions to pathologies is challenging due to the difficulty in accounting for environmental influences and disease comorbidities. Underlying epistatic interactions may also contribute to problems with replication in human GWAS performed in distinct populations which rarely take epistatic effects into account. In addition, linking a trait associated locus to a candidate gene or a set of genes for prioritization is not straightforward (*Mackay, 2014*, *Boyle et al., 2017*). Furthermore, the analysis of genetic factors related to cardiac traits is complicated by their interactions with several risk factors, such as increasing age, hypertension, diabetes mellitus, ischemic, and structural heart disease (*Paludan-Müller et al., 2016*).

These pitfalls can be overcome using animal models. Model organisms allow precise controlling of the genetic background and environmental rearing conditions. They can provide generally applicable insights into the genetic underpinnings of complex traits and human diseases, due to the evolutionary conservation of biological pathways. Numerous studies have highlighted the conservation of cardiac development and function from flies to mammals. Indeed, orthologous genes control the early development as well as the essential functional elements of the heart. The fly is the simplest genetic model with a heart muscle and is increasingly used to identify the genes involved in heart disease and aging (*Ocorr et al., 2007b*; *Diop and Bodmer, 2015*; *Rosenthal et al., 2010*). Although a large number of genes are implicated in establishing and maintaining cardiac function in *Drosophila* (*Neely et al., 2010*), the extent to which genes identified from mutant analysis reflect naturally occurring variants is neither known, nor do we know how allelic variants at several segregating loci combine to affect cardiac performance.

We previously showed that wild populations of flies harbor rare polymorphisms of major effects that predispose them to cardiac dysfunction (*Ocorr et al., 2007a*). Here, we analyzed the genetic architecture of the natural variation of cardiac performance in *Drosophila*. Our aims were to (i) identify the variants associated with cardiac traits found in a natural population, (ii) decipher how these variants interact with each other and with the environment to impact cardiac performance, and (iii) gain insights into the molecular and cellular processes affected. For this, we used the *Drosophila* Genetic Reference Panel (DGRP) (*Mackay et al., 2012*; *Huang et al., 2014*), a community resource of sequenced inbred lines. Previous GWAS performed in the DGRP indicate that inheritance of most quantitative traits in *Drosophila* is complex, involving many genes with small additive effects as well as epistatic interactions (*Mackay and Huang, 2018*). The use of inbred lines allows us to assess the effects of genetic variations in distinct but constant genetic backgrounds and discriminate genetic and environmental effects.

We demonstrated substantial among-lines variations of cardiac performance and identified genetic variants associated with the cardiac traits together with epistatic interactions among polymorphisms. Candidate loci were enriched for genes encoding transcription factors (TFs) and signaling pathways, which we validated in vivo. We used non-coding variants - which represented the vast majority of identified polymorphisms – for predicting transcriptional regulators of associated genes. Corresponding TFs were further validated in vivo by heart-specific RNAi-mediated knockdown (KD). This illustrates that natural variations of gene regulatory networks have widespread impact on cardiac function. In addition, we analyzed the phenotypic variability of cardiac traits between individuals within each of the DGRP lines (i.e., with the same genotype), and we documented significant diversity in phenotypic variability among the DGRP lines, suggesting genetic variations influenced phenotypic variability of cardiac performance. Genetic variants associated with this phenotypic variability were identified and shown to affect a set of genes that overlapped with those associated with trait means, although through different genetic variants in the same genes.

Comparison of human GWAS of cardiac disorders with results in flies identified a set of orthologous genes associated with cardiac traits both in *Drosophila* and in humans, supporting the conservation of the genetic architecture of cardiac performance traits, from arthropods to mammals. siRNA-mediated gene KD were performed in human induced pluripotent stem cells derived cardiomyocytes (hiPSC-CMs) to indeed show that *dmPox-meso*/hPAX9 and *dmStripe*/hEGR2 have conserved functions in cardiomyocytes from both flies and humans. These new insights into the fly's genetic architecture and the connections between natural variations and cardiac performance permit the accelerated identification of essential cardiac genes and pathways in humans.

# Results

## Quantitative genetics of heart performances in the DGRP

In this study, we aimed to evaluate how naturally occurring genetic variations affect cardiac performance in young *Drosophila* adults and identify variants and genes involved in the genetic architecture of cardiac traits. To assess the magnitude of naturally occurring variations of the traits, we measured heart parameters in 1-week-old females for 167 lines from the DGRP, a publicly available population of sequenced inbred lines derived from wild caught females (*Figure 1A*). Briefly, semi-intact preparations of individual flies (*Ocorr et al., 2007c*) were used for high-speed video recording combined with Semi-automated Heartbeat Analysis (SOHA) software (http://www.sohasoftware.com/) which allows precise quantification of a number of cardiac function parameters (*Fink et al., 2009*; *Cammarato et al., 2015*). Fly cardiac function parameters are highly influenced by sex (*Wessells et al., 2004*). Due to the experimental burden of analyzing individual cardiac phenotypes, we focused on female flies only and designed our experiment in the following way: we randomly selected 14 lines out of 167 that were replicated twice. The remaining 153 lines were replicated once. Each replicate was composed of 12 individuals. No block effect was observed due to the replicates in the 14 selected lines (see *Supplementary file 1a*). This allowed us to perform our final analysis on one replicate of each of the 167 lines. A total sample of 1956 individuals was analyzed. Seven cardiac traits were analyzed across the whole population (*Figure 1—source data 1* and *Table 1*). As illustrated in *Figure 1A*, we analyzed phenotypes related to the rhythmicity of cardiac function: the systolic interval (SI) is the time elapsed between the beginning and the end of one contraction, the diastolic interval (DI) is the time elapsed between two contractions and the heart period (HP) is the duration of a total cycle (contraction+relaxation (DI+SI)). The arrhythmia index (AI, std-dev(HP)/mean (HP)) is used to evaluate the variability of the cardiac rhythm. In addition, three traits related to contractility were measured. The diameters of the heart in diastole (end diastolic diameter [EDD]), in systole (end systolic diameter [ESD]), and the fractional shortening (FS), which measures the contraction efficacy (EDD −ESD/EDD). We found significant genetic variation for all traits (*Figure 1B* and *Figure 1—figure supplement 1*) with broad sense heritability ranging from 0.30 (AI) to 0.56 (EDD) (*Table 1*). Except for EDD/ESD and HP/DI, quantitative traits were poorly correlated with each other (*Figure 1—figure supplement 1*).

## GWAS analyses of heart performance

To identify candidate variants associated with cardiac performance variation, we performed GWAS analyses and evaluated single marker associations of line means with common variants using a linear mixed model (*Lippert et al., 2011*) and after accounting for effects of Wolbachia infection and common polymorphism inversions (see Materials and methods). Genotype-phenotype associations were performed separately for all seven quantitative traits and variants were ranked based on their p-values. For most of the phenotypes analyzed, quantile-quantile (QQ) plots were uniform (*Figure 1—figure supplement 2*) and none of the variants reached the strict Bonferroni correction threshold for multiple tests ($2 \cdot 10^{-8}$), which is usual in the DGRP given the size of the population. However, the decisive advantage of the *Drosophila* system is that we can use GWA analyses as primary screens for candidate genes and mechanisms that can be subsequently validated by different means. We therefore chose to analyze the 100 top ranked variants for each quantitative trait. This choice is based on our strategy to test the selected single nucleotide polymorphisms (SNPs) and associated genes by a variety of approaches – data mining and experimental validation (see below) – in order to provide a global validation of association results and to gain insights into the characteristics of the genetic architecture of the cardiac traits. This cut-off was chosen in order to be able to test a significant number of variants while being globally similar to the nominal cut-off ($10^{-5}$) generally used in DGRP analyses. A large proportion of the variants retained have indeed a p-value below $10^{-5}$. Selected variants were further filtered on the basis of minor allele frequency (MAF >4%) (*Figure 1—source data 2*, *Supplementary file 1b*). Among the seven quantitative traits analyzed, we identified 530 unique variants. These variants were associated to genes if they were within 1 kb of transcription start site (TSS) or transcription end sites (TES). Using these criteria, 417 variants were mapped to 332 genes (*Supplementary file 1c*). We performed a chi-squared test to determine if the genomic location of variants associated with cardiac traits is biased toward any particular genomic region when compared with the whole set of variants with MAF >4% in the DGRP population and obtained a p-value of 2.778e-13. Genomic locations of the variants were

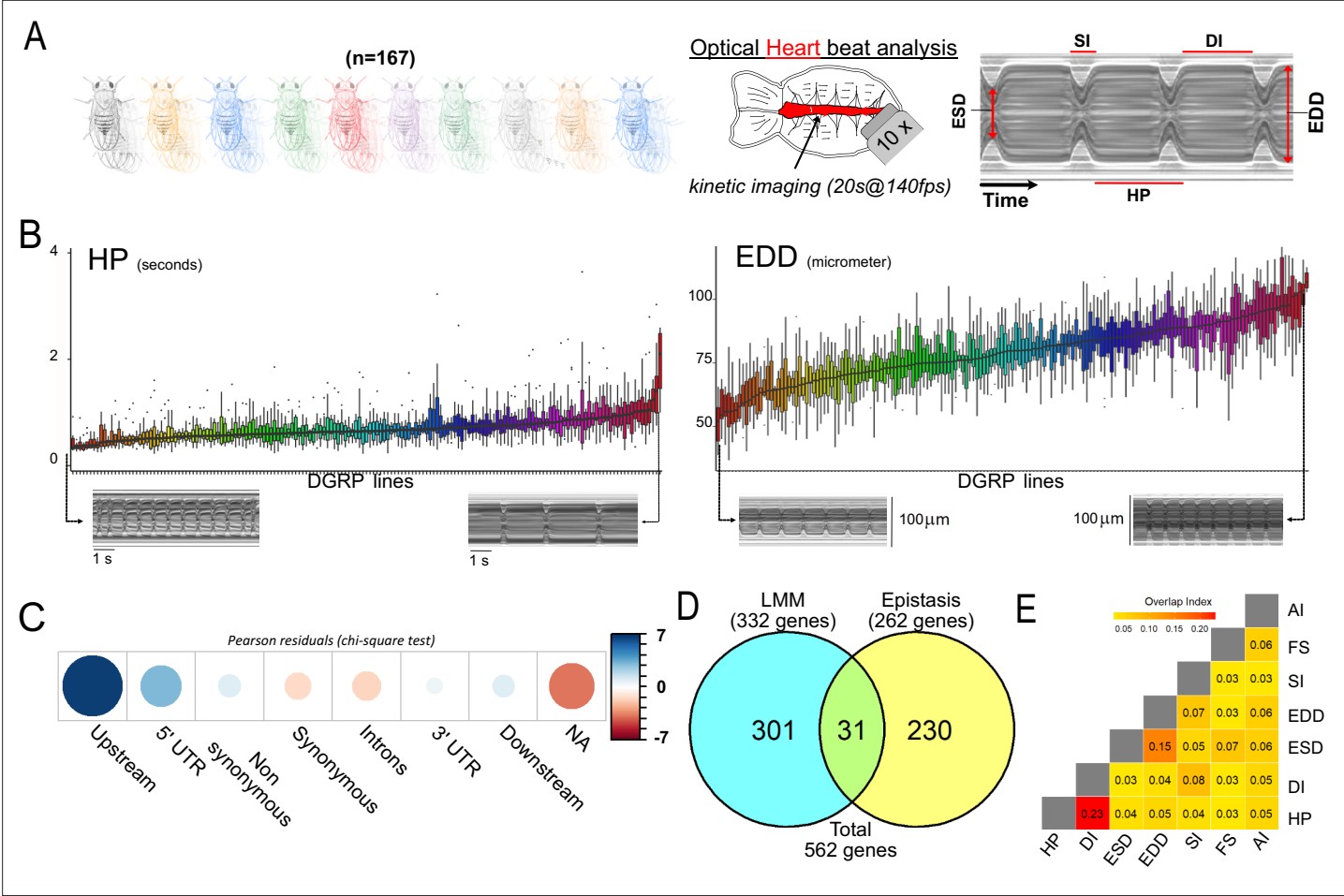

**Figure 1.** Quantitative genetics and genome-wide associations studies (GWAS) for cardiac traits in the *Drosophila* Genetic Reference Panel (DGRP).
(**A**) Left: Cardiac performance traits were analyzed in 167 sequenced inbred lines from the DGRP population. Approximately 12 females per line were analyzed. Right panels: Schematic of the *Drosophila* adult heart assay and example of M-mode generated from video recording of a beating fly heart. Semi-intact preparations of 1-week-old adult females were used for high-speed video recording followed by automated and quantitative assessment of heart size and function. The representative M-mode trace illustrate the cardiac traits analyzed. DI: diastolic interval; SI: systolic interval; HP: heart period (duration of one heartbeat); EDD: end diastolic diameter (fully relaxed cardiac tube diameter); ESD: end systolic diameter (fully contracted cardiac tube diameter). Fractional shortening (FS=EDD − ESD/EDD) and arrythmia index (AI=Std Dev (HP)/HP) were additionally calculated and analyzed.
(**B**) Distribution of line means and within lines variations (box plots) from 167 measured DGRP lines for HP and EDD. DGRP lines are ranked by their increasing mean phenotypic values. For both phenotypes, representative M-modes from extreme lines are shown below (other traits are displayed in *Figure 1—figure supplement 1*). (**C**) Pearson residuals of chi-square test from the comparison of indicated single nucleotide polymorphism (SNP) categories in the DGRP and among variants associated with cardiac traits. According to DGRP annotations, SNPs are attributed to genes if they are within the gene transcription unit (5' and 3' UTR, synonymous and non-synonymous coding, introns) or within 1 kb from transcription start and end sites (1 kb upstream, 1 kb downstream). NA: SNPs not attributed to genes (>1 kb from transcription start site [TSS] and transcription end sites [TES]). (**D**) Comparison of gene sets identified by single marker using Fast-LMM (LMM) and in interaction using FastEpistasis (Epistasis). The Venn diagram illustrates the size of the two populations and their overlap. (**E**) Overlap coefficient of gene sets associated with the different cardiac traits analyzed.

The online version of this article includes the following source data and figure supplement(s) for figure 1:

**Source data 1.** Individual values for cardiac traits analyzed across the 167 *Drosophila* Genetic Reference Panel (DGRP) lines.

**Source data 2.** Variants identified by FastLMM as associated to indicated phenotypes.

**Source data 3.** All FastEpistasis data on mean phenotypes, per quantitative trait.

**Figure supplement 1.** Quantitative genetics of cardiac traits means.

**Figure supplement 2.** Quantile-quantile (QQ) plots of association p-values.

**Table 1.** Quantitative genetics of cardiac traits in the *Drosophila* Genetic Reference Panel (DGRP).
Summary statistics over all DGRP genotypes assayed. Number of lines and individuals (after outlier removal, see Materials and methods) analyzed for each cardiac trait is indicated. Mean, standard deviation (Std dev.), and coefficient of variation (Coef. Var) among the whole population are indicated. Genetic, environment, and phenotypic variance (respectively Genet. var, Env. var, and Phen. var) were calculated for each trait. Broad sense heritability of traits means (H2) suggested heritability of corresponding traits. Levene test indicated significant heterogeneity of the variance among the lines.

| | Diastolic intervals | Systolic intervals | Heart period | Diastolic Diameter | Systolic diameter | Fractional shortening | Arrhythmia Index |
|---|---|---|---|---|---|---|---|
| total.nb.lines | 167 | 167 | 167 | 167 | 167 | 167 | 167 |
| mean | 0.4638 | 0.2166 | 0.6883 | 79.4200 | 51.0500 | 0.3538 | 0.2475 |
| Std dev. | 0.26330 | 0.03216 | 0.27690 | 14.09000 | 9.49300 | 0.06837 | 0.29230 |
| Coef. var | 0.5677 | 0.1485 | 0.4022 | 0.1774 | 0.1860 | 0.1933 | 1.1810 |
| lines (mean) | 165 | 166 | 165 | 159 | 157 | 158 | 166 |
| Indiv. (mean) | 1914 | 1911 | 1920 | 1779 | 1753 | 1767 | 1832 |
| lines (Cve) | 165 | 166 | 165 | 159 | 157 | 158 | 166 |
| Indiv. (Cve) | 1914 | 1911 | 1920 | 1779 | 1753 | 1767 | 1832 |
| Genet. var | 2.59e-02 | 5.03e-04 | 2.87e-02 | 1.13e+02 | 4.39e+01 | 1.57e-03 | 2.21e-02 |
| Env. var | 4.36e-02 | 5.35e-04 | 4.82e-02 | 8.64e+01 | 4.65e+01 | 3.11e-03 | 6.35e-02 |
| Phen. var | 6.95e-02 | 1.04e-03 | 7.68e-02 | 1.99e+02 | 9.04e+01 | 4.68e-03 | 8.56e-02 |
| H2 | 0.373 | 0.485 | 0.373 | 0.566 | 0.485 | 0.335 | 0.258 |
| F value | 76,864 | 11,686 | 74,715 | 46,950 | 15,041 | 11,164 | 65,308 |
| Pr(F) | 8.8e-120 | 2.3e-187 | 5.8e-116 | 7.1e-62 | 1.9e-231 | 8.8e-175 | 1.8e-96 |
| Levene test | 1.9e-10 | 1.9e-10 | 1.7e-08 | 1.6e-05 | 2.1e-13 | 1.6e-05 | 2.1e-13 |

biased toward regions within 1 kb upstream of genes TSS, and, to a lesser extent, to genes 5' UTR (*Figure 1C*). Variants not mapped to genes (located at >1 kb from TSS or TES) were slightly depleted in our set.

In GWAS analyses, loci associated with a complex trait collectively account for only a small proportion of the observed genetic variation (*Manolio et al., 2009*) and part of this 'missing heritability' is thought to come from interactions between variants (*Flint and Mackay, 2009*; *Manolio et al., 2009*; *Huang et al., 2012*; *Shorter et al., 2015*). As a first step toward identifying such interactions, we used FastEpistasis (*Schüpbach et al., 2010*). SNP identified by GWAS were used as focal SNPs and were tested for interactions with all other SNPs in the DGRP. FastEpistasis reports best ranked interacting SNP for each starting focal SNP, thus extending the network of variants and genes associated to natural variation of cardiac performance, which were used for hypothesis generation and functional validations; 288 unique SNPs were identified, which were mapped to 261 genes (*Figure 1—source data 3*, *Supplementary file 1e*). While none of the focal SNPs interacted with each other, there is a significant overlap between the 332 genes associated with single marker GWAS and the 261 genes identified by epistasis (n=31, *Figure 1D* and *Supplementary file 1e*, fold change (FC)=6; hypergeometric pval=6.8 × 10⁻¹⁶). This illustrates that the genes that contribute to quantitative variations in cardiac performance have a tendency to interact with each other, although through distinct alleles.

Taken together, single marker GWAS and epistatic interactions performed on the seven cardiac phenotypes identified a compendium of 562 genes associated with natural variations of heart performance (*Supplementary file 1f*). In line with the correlation noted between their phenotypes (*Figure 1—figure supplement 1B*), the GWAS for HP and DI identified partially overlapping gene sets (overlap index 0.23, *Figure 1E*). The same was true, to a lesser extent, for ESD and EDD (0.15). Otherwise, the sets of genes associated with each of the cardiac traits are poorly correlated with each other.

## Functional annotations and network analyses of association results

Our next objective was to identify the biological processes potentially affected by natural variation in cardiac performance. Gene Ontology (GO) enrichment analysis of the combined single marker GWAS and epistatic interactions analyses indicated that genes encoding signaling receptors, TFs, and cell adhesion molecules were over-represented among these gene sets (pval=1.4 × 10$^{-9}$ [FC=2.9], 5×10$^{-4}$ [FC=2], and 3×10$^{-3}$ [FC=4.6], respectively). There was also a bias for genes encoding proteins located at the plasma membrane, at ion channel complexes as well as components of contractile fibers (pval=3.4 × 10$^{-10}$ [FC=3], 7×10$^{-5}$ [FC=4.2], and 4×10$^{-2}$ [FC=3.6]; *Figure 2A*; *Supplementary file 2a*). Of note, although a number of genes have previously been identified as being required during heart development or for the establishment and maintenance of cardiac function by single gene approaches, we found no enrichment for these gene categories in our analysis. In addition, genes identified in a global screen for stress-induced lethality following heart-specific RNAi KD (*Neely et al., 2010*) were also not enriched in GWAS detected genes (FC=1; *Supplementary file 2b*). This indicates that genes associated to natural variations of cardiac traits are typically missed by traditional forward or reverse genetics approaches, which highlights the value of our approach.

In order to gain additional knowledge about the cellular and molecular pathways affected by natural variations of cardiac traits, we have mapped the associated genes and gene products onto characterized interaction networks. Of the 562 identified genes, 419 were mapped to the fly interactome that includes both physical (protein-protein) and genetic interactions from both DroID (*Murali et al., 2011*) and flybi databases (see Materials and methods and *Figure 2—source data 1*). Remarkably, a high proportion (148) of the GWAS identified genes were directly connected within the fly interactome and formed a large network of interacting genes/proteins (*Supplementary file 2c and d*, *Figure 2B*), suggesting that they participate in common biological processes. This network encompasses several TFs and ion channel complex genes, consistent with their potential role in the genetic architecture of natural variation of heart performance. Several components of signaling pathways are also present in the network, including members of the FGF and TGFβ pathways (see below).

## Functional validations of candidate genes

To assess in an extensive way whether mutations in genes harboring SNPs associated with variation in cardiac traits contributed to these phenotypes, we selected 42 GWAS associated genes for cardiac-specific RNAi KD and tested the effects on cardiac performance. We selected genes that were identified in at least two independent GWAS for two traits or that were known to be dynamically expressed in the adult heart (*Monnier et al., 2012*) and for which inducible RNAi lines were available. Genes were tested in 1-week-old adult female flies, using the heart-specific Hand>Gal4 driver (*Popichenko et al., 2007*) and the same semi-intact heart preparations and SOHA analyses as for DGRP lines screening. Notably, 38 of the 42 selected genes led to various levels of cardiac performance defects following heart-specific KD (*Figure 2C*). In parallel, we tested the effect on cardiac performance of knocking down 18 genes randomly selected in the genome – the GWAS associated genes being excluded from the selection (see Materials and methods and *Figure 2—figure supplement 1*). Although a number of these genes lead to cardiac phenotypes when inactivated – which is consistent with published observations that quantitative traits can be influenced by a large number of genes (*Zhang et al., 2021*) – when inactivated in the heart, the genes selected from GWAS lead to significantly more frequent phenotypes compared to the randomly selected genes (*Figure 2—figure supplement 1*). These results therefore supported our association results. It is important to emphasize that our approach is limited to testing the effects of tissue-specific gene KD. Since some of the variants may lead to increased gene function and/or expression, this can lead to a false negative rate that is difficult to estimate. In addition, some of the associated variants may influence heart function by non-cell-autonomous mechanisms, which would not be replicated by cardiac-specific RNAi KD.

We further focused on the TGFβ pathway, since members of both BMP and activin pathways were identified in our analyses. We tested different members of the TGFβ pathway for cardiac phenotypes using cardiac-specific RNAi KD (*Figure 2C*), and confirmed the involvement of the activin agonist *snoo* (Ski orthologue) and the BMP antagonist *sog* (chordin orthologue). Notably, activin and BMP pathways are usually antagonistic (*Figure 2D*). Their joint identification in our GWAS suggest that they act in a coordinated fashion to regulate heart function. Alternatively, it may simply reflect their involvement in different aspects of cardiac development and/or functional maturation. In order to discriminate

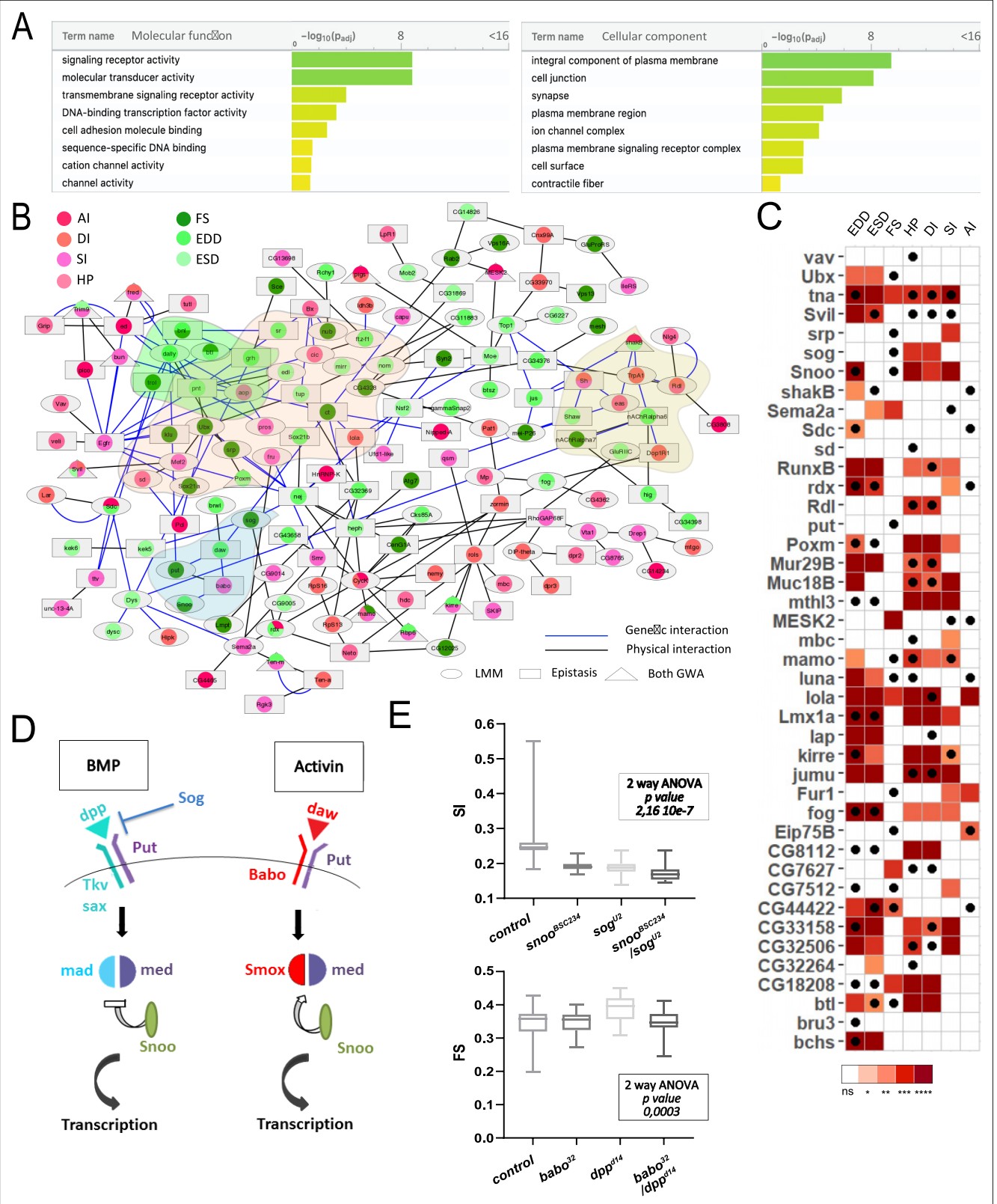

**Figure 2.** Functional annotations and validations of genes associated with genome-wide associations studies (GWAS) for cardiac performance. (**A**) Gene Ontology (GO) enrichment analyses. Selected molecular functions (MF, left) and cellular components (CC, right) associated with cardiac performances at FDR < 0.05 are shown. Enrichment analysis was performed using G:profiler with a correction for multitesting (see Materials and methods). (**B**) Interaction network of genes associated with natural variations of cardiac performance. Direct genetic and physical interactions between cardiac fly GWAS genes

*Figure 2 continued on next page*

*Figure 2 continued*

are displayed. Nodes represent genes and/or proteins, edges represent interactions (blue: genetic; black: physical). Node shapes refer to single marker and/or epistasis GWAS, node color to the cardiac performances phenotype(s) for which associations were established. Genes and proteins highlighted in pink point to transcription factors, in green and blue to signaling pathways (FGF and TGFb, respectively), and in yellow to ion channels. (**C**) Heatmap representing the effects on indicated cardiac traits of heart-specific RNAi-mediated knockdown (KD) of 42 genes identified in GWAS for cardiac performance. Results of Wilcoxon rank sum test of the effects of indicated heart-specific RNAi-mediated gene KD (rows) for cardiac performance traits (columns), analyzed on semi-intact 1-week females flies. Detailed data are presented in *Figure 2—source data 2*. Thirty-eight (out of 42) genes tested lead to significant effects on cardiac performance traits upon KD. Black dots indicate the trait(s) for which the corresponding gene was associated in GWAS. ns: not significant; *: pval <0.05; **: pval <0.01; ***: pval <0.005; ****pval <0.0001 (p-values were adjusted for multiple testing using Bonferroni correction). Comparison with heart-specific effect of random selected genes is displayed in *Figure 2—figure supplement 1*. (**D**) Schematic drawing of BMP and activin pathways in *Drosophila*. (**E**) Genetic interactions between BMP and activin pathway genes. Genetic interactions tested between *snoo^BSC234^* and *sog^U2^* for SI and between *dpp^d14^* and *babo^32^* for FS (other phenotypes are shown in *Figure 2—figure supplement 2*). Cardiac traits were measured on each single heterozygotes and on double heterozygotes flies. Two-way ANOVA reveals that the interaction between *snoo^BSC234^* and *sog^U2^* for SI and between *dpp^d14^* and *babo^32^* for FS are significant. Detailed data for interaction effect corresponding to all phenotypes are displayed in *Figure 2—figure supplement 2*.

The online version of this article includes the following source data and figure supplement(s) for figure 2:

**Source data 1.** Collection of physical (IP, Y2H) and genetic interactions identified in *Drosophila*.

**Source data 2.** Data from validation experiments.

**Figure supplement 1.** Heart-specific RNAi-mediated knockdown (KD) of random selected genes.

**Figure supplement 2.** Genetic interactions between BMP and activin pathway genes.

between these two hypotheses, we tested if different components of these pathways interacted genetically. Single heterozygotes for loss of function alleles show dosage-dependent effects of *snoo* and *sog* on several phenotypes, providing an independent confirmation of their involvement in several cardiac traits (*Figure 2—figure supplement 2*). Importantly, compared to each single heterozygotes, *snoo^BSC234^/sogU^U2^* double heterozygotes flies showed non-additive SI phenotypes (two-way ANOVA pval: $2.1 \cdot 10^{-7}$), suggesting a genetic interaction (*Figure 2E* and *Figure 2—figure supplement 2*). It is worth noting however that *snoo* is also a transcriptional repressor of the BMP pathway (*Takaesu et al., 2006*). The effect observed in *snoo^BSC234^/sogU^U2^* double heterozygotes can therefore alternatively arise as a consequence of an increased BMP signaling without affecting the activin pathway. We thus tested other allelic combinations for loss of function alleles of BMP and activin pathways. *babo/tkv* heterozygotes (respectively activin and BMP type 1 receptors) displayed non-additive ESD and EDD phenotypes (*Figure 2—figure supplement 2*). Synergistic interaction of BMP and activin pathways was also suggested by the analysis of FS in loss of function mutants for *babo* and *dpp*, the BMP ligand (*Figure 2—figure supplement 2*). Of note, *babo/tkv* double heterozygotes also displayed a tendency to non-additive effects on FS albeit non-significant (two-way ANOVA p=0.054). In addition, *mad/smox* heterozygotes (specifc downstream TFs of BMP and activin pathways) displayed non-additive effects on several traits, including phenotypes related to rhythmicity (HP, SI, DI) and contractility (ESD and EDD) (*Figure 2—figure supplement 2*). Altogether, cardiac performance in response to allelic combinations of activin and BMP supported a coordinated action of both pathways in the establishment and/or maintenance of cardiac activity. This was further supported by the observation that simple heterozygotes for the tested loss of function alleles displayed similar trends with respect to cardiac performance, irrespective of the pathway considered (*dpp, tkv,* and *mad* for BMP; *babo* and *smox* for activin). Indeed, they displayed either no effect or increased FS and rhythmicity phenotypes (HP, DI, SI, AI), and decreased cardiac diameters (ESD and EDD). This suggests coordinated activity of both pathways. Importantly, the genetic interactions were tested using amorphic alleles that lead to systemic loss of function. The observed phenotypes may thus not unravel cardiac-specific effects of the pathways. In support of this, *mad* cardiac-specific RNAi KD was tested (see below, *Figure 3*) and lead to a decreased HP, DI, SI, and FS whereas heterozygotes for *mad Neely et al., 2010* have either no (FS) or opposite (HP, DI, SI) effect on these phenotypes (*Figure 2—figure supplement 2*). Inversely, *mad* RNAi caused a significant increase in AI whereas *mad* (*Neely et al., 2010*) had no effect. However, heart-specific *dpp* RNAi KD (*Figure 2—figure supplement 2*) lead to similar phenotypic trends compared to *dpp^d14^* (increased HP, DI, SI, decreased EDD and ESD) with the notable exception of FS which was reduced following cardiac-specific KD (*Figure 2—figure supplement 2*), but increased in *dpp^d14^* heterozygotes (*Figure 2—figure supplement 2*). Taken together, these data

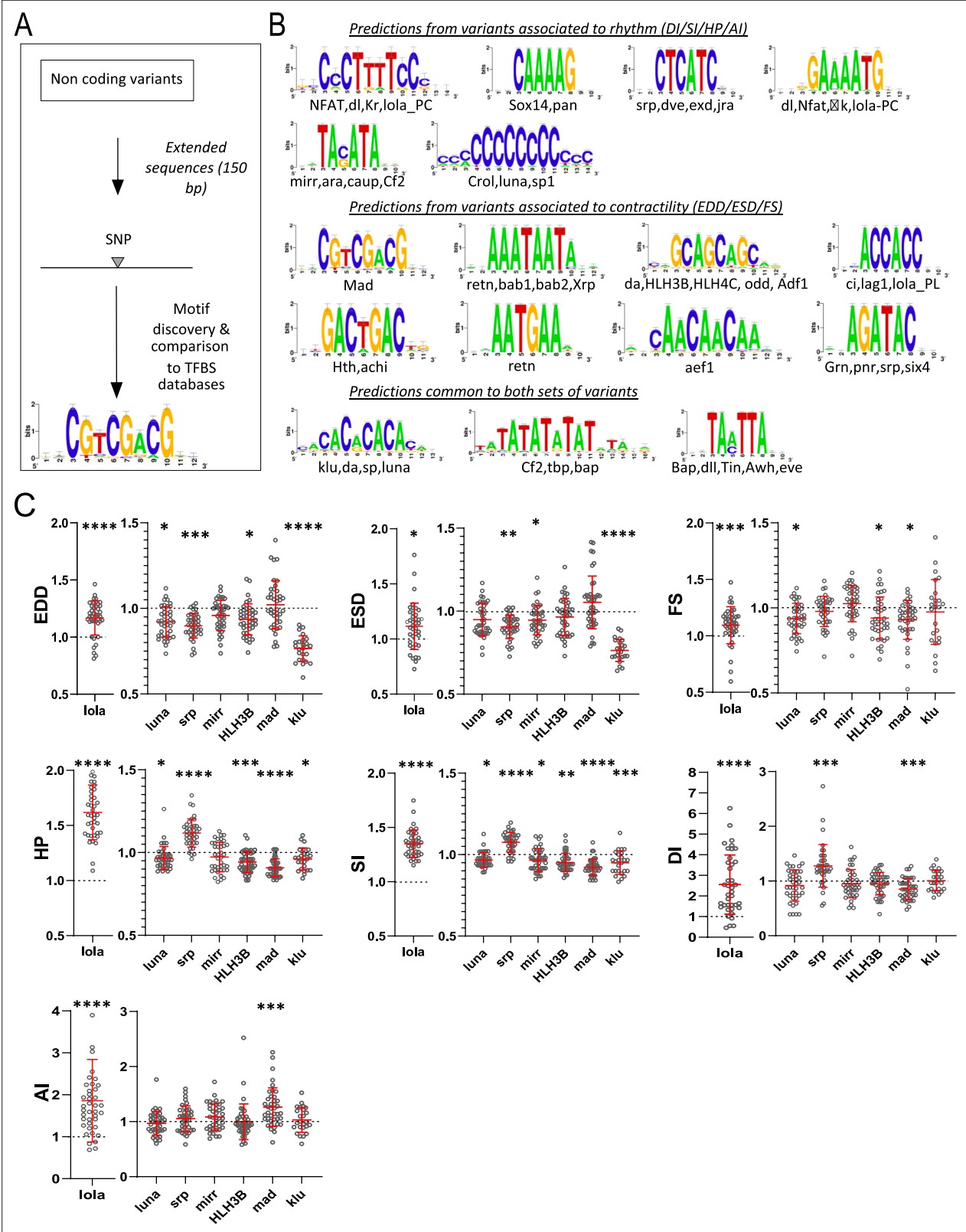

**Figure 3.** Transcription factor binding site (TFBS) predictions from sequences surrounding non-coding variants and in vivo validation of cognate transcription factors (TFs). (**A**) Schematic representation of the workflow used for motifs analysis in the vicinity of the non-coding variants and comparison with TFBSs databases. (**B**) Logo representation of position-specific scoring matrices (PSSMs) identified in corresponding sets of non-coding variants. Potential cognate TFs are indicated and were identified by comparing PSSMs to TFBSs databases. (**C**) Cardiac-specific RNAi knockdown

*Figure 3 continued on next page*

Figure 3 continued

phenotypes of TFs tested in vivo. Hand>Gal4; UAS RNAi 1-week adult females (n>30 per cross) were analyzed for cardiac performance in intact flies. Scatter plots of individual data, mean, and SD are represented. Wilcoxon test results are indicated (*: pval <0.05; **: pval <0.01; ***: pval <0.005; ****pval <0.0001). Data were normalized to genotype matched controls. Raw data are presented in *Figure 2—source data 2*. Detailed statistical analyses are in *Supplementary file 2e*.

The online version of this article includes the following source data for figure 3:

**Source data 1.** Variants in non-coding regions used for motif discovery, associated with end diastolic diameter (EDD), end systolic diameter (ESD), and fractional shortening (FS) ('structure') and to AI, DI, SI, and HP ('rhythm') for both mean and CVe of cardiac traits.

point to a complex picture of TGFβ pathway activity in regulating cardiac performance, involving both the activin and the BMP pathways as well as gene-specific effects with both systemic and tissue-specific contributions.

As a whole, the cardiac phenotypes observed following RNAi-mediated gene KD of GWAS selected genes and the complex interplay characterized between different activin and BMP pathways members are strong validations of the GWAS results that are further supported by the functional analysis of non-coding variants and by the correlations we observed between identified fly genes and human genes associated to cardiac dysfunction (discussed below).

## Natural variations of heart performance primarily affect gene regulatory networks

The over-representation of genes coding for TFs found in GWAS for cardiac performance traits (pval=5559 · 10$^{-4}$, *Figure 2A*, *Supplementary file 2a*) suggested that perturbations of gene regulatory networks (GRNs) play a central role in the genetic architecture of natural variations of cardiac performance. In addition, most variants (95%) were localized in non-coding regions of the genome. Remarkably, their distribution was enriched in gene proximal (0–1 kb) upstream regions (*Figure 1C*) suggesting that a high proportion of variants affects cis-regulatory sequences.

Thus, we sought to determine whether potential TF binding motifs are enriched in the vicinity of variants identified by GWAS. Analysis of the sites of the variants themselves did not reveal any particular pattern. However, oligo analysis (*van Helden et al., 1998*) of the 150 bp sequences around these non-coding variants detected significant over-represented k-mers (Evalue ≤ 1e-4). These were further assembled and converted into positions-specific scoring matrices (PSSMs) and compared with PSSM databases of known fly TFs binding motifs were used for annotations (see Materials and methods, *Figure 3—source data 1* and *Figure 3A*). Variants associated with quantitative traits linked to heart size and contractility (ESD, EDD, and FS; 367 variants) and those related to heart rhythmicity (AI, HP, DI, SI; 435 variants) were analyzed separately. We identified enrichment of potential DNA binding motifs for both sets of traits (*Figure 3B*). Although this approach predicted motifs that were specific for each group of variants, a significant proportion of potential DNA binding motifs were predicted by both groups, suggesting commonalities in the genetic architecture of the corresponding traits.

Remarkably, we identified DNA binding motifs for TFs that were themselves associated with natural variations of the cardiac traits. Some examples are *klumpfuss (klu/ZBTB7A)*, *luna (KLF5-7)*, *longitudinals lacking (lola/BTB18)*, *serpent (srp/GATA1-6)*, *mirror (mirr/IRX3-6)*, *Helix Loop Helix protein 3B (HLH3B/TAL2)*, and *defective proventiculus (dve/SATB1)* which were also identified by GWAS (see *Supplementary file 1f*). Potential binding sites for mothers against dpp, Mad (SMAD) were enriched in the vicinity of the non-coding variants associated with heart size and contractility. SMAD is the downstream effector of the BMP signaling pathway that we showed was also involved in regulating cardiac contractility (*Figure 2D–E*). Motifs predicted to be bound by the NK TFs Tinman (Tin/NKX2.5) and Bagpipe (Bap/NKX3.2) are over-represented in both sets of variants. *tin* has a well-characterized function in the cardiac GRN, both during embryogenesis (*Bodmer, 1993*) and at adulthood (*Qian et al., 2011*) and *bap* is also a member of the cardiac network (*Seyres et al., 2016*). Lastly, we found *Jun*-related antigen (Jra/JUN) TF binding sites (TFBS) enriched in sequences associated with natural variations of cardiac rhythm. We previously reported that *djun* KD leads to altered HP and AI in young flies (*Monnier et al., 2012*), supporting a function for this stress-activated TF in the cardiac GRN.

We did not test individually the effects on cardiac performance of mutations in predicted TFBSs located near the SNPs because any individual effect would probably be too small to be detectable by the available methods. Rather, we tested the potential involvement of their cognate TFs by

cardiac-specific RNAi-mediated KD. *luna*, *lola*, and *srp* KD had already been analyzed in semi-intact preparations and shown to induce strong cardiac phenotypes (*Figure 2C*). We repeated these heart-specific manipulations employing additional, independent RNAi lines targeting these genes and an in vivo approach that uses the red fluorophore TdTomato driven by a cardiac enhancer (*Klassen et al., 2017*) to monitor heart wall movements. To validate roles for the predicted TFBSs, we also performed heart-specific KD of *mirr*, *mad*, *klu,* and *HLH3B* using this same in vivo approach. As illustrated in *Figure 3C*, all tested TFs are associated with changes in multiple cardiac traits. *mad* KD affected FS but also impacted several traits related to rhythmicity. Consistent with the predictions of its binding sites, *klu* KD altered HP and SI but also EDD and ESD. A PSSM for one isoform of *lola* (Lola-RC) was predicted from variants associated with rhythmicity, and another one (for Lola-RL) from variants related to contractility. Of note, *lola* KD strongly impacted all analyzed cardiac phenotypes. *luna* cardiac-specific RNAi KD had an impact on EDD, FS, HP, and SI, in line with its binding sites predictions. Similarly, *srp* KD impacted HP, DI, and SI and also modified cardiac diameters. Lastly, *mirr* KD affected both SI and ESD and *HLH3B* impacted EDD, HP, and SI (*Figure 3C* and *Supplementary file 2e*).

Taken together, the high rate of non-coding SNPs among variants identified in GWAS, the TFBS predictions with in vivo validation of potential cognate TFs, and enrichment for TFs among genes associated with natural variations of heart performance support a central role of GRN deviations in the natural variation of cardiac performance.

## Identifying natural variations of phenotypic variability of cardiac performance

The use of inbred lines offers the possibility of studying the heritability of the within-line variance of the phenotypes. Several studies have indeed shown that the within-line environmental variance of inbred lines is heritable (*Morgante et al., 2015*; *Harbison et al., 2013*; *Ayroles et al., 2015*). These genetic variations in phenotypic variability suggest that different genotypes respond differently in response to micro-environmental changes (*Geiler-Samerotte et al., 2013*; *Anholt and Mackay, 2018*). We used the Levene test *Levene, 1960* to examine heterogeneity of the 'micro-environmental' variance among lines for the seven analyzed cardiac traits. We found evidence of significant heterogeneity among the lines in the variance for all analyzed cardiac traits (*Table 1*). This suggests that the lines differ in their ability to adapt to micro-environmental variations, and supports the heritability of this differing adaptability. The coefficient of within-line variation (CVe) was used as the metric for within-line environmental variance. We found significant genetic variation for all CVe traits (*Figure 4A and B* and *Figure 4—figure supplement 1A*) indicating that within some lines, individuals display relatively constant cardiac performance traits, whereas within others, individuals display widely varying cardiac traits. Similar to trait means, CVe traits were poorly correlated with each other (*Figure 4—figure supplement 1B*), except for HP/DI and EDD/ESD. Correlations of CVe traits with their respective trait means ranged from moderate (FS: –0.53) to none (DI: 0.064, SI: 0.09, ESD: 0.08) (*Figure 4—figure supplement 1C*). This suggested that distinct loci may affect natural variations of CVe and means of cardiac performance parameters.

We performed GWAS analysis to identify candidate variants and genes associated with variation of CVe of cardiac performance using both single marker and epistasis GWAS analyses. Similar to trait means we retained the 100 top ranked variants for reporting single marker associations. These variants were further used as focal variants for epistasis detection and the best ranked interacting SNP for each starting focal SNP was identified. Overall, this led to the identification of 887 variants and 566 genes associated with natural variations of CVe of cardiac performance (*Figure 4—source data 1*, *Supplementary file 3a*), which were used for hypothesis generation and functional annotations. Although the number of individuals we were able to analyze – due to the experimental burden of analyzing individual cardiac phenotypes – reduced the statistical power of the analysis of micro-environmental variance, the functional enrichment analyses of variants and genes described below provide support for our association results and highlight important features of the genetic architecture of cardiac traits.

Consistent with the weak correlation between cardiac trait CVe and trait means, we found only three variants associated with both (*Figure 4C and D* and *Figure 4—figure supplement 2*; *Supplementary file 3c*). Two variants in total linkage disequilibrium (LD) (5 bp apart) are localized in an intron of *Ca-β*, encoding a voltage gated calcium channel subunit, and are associated for both mean and

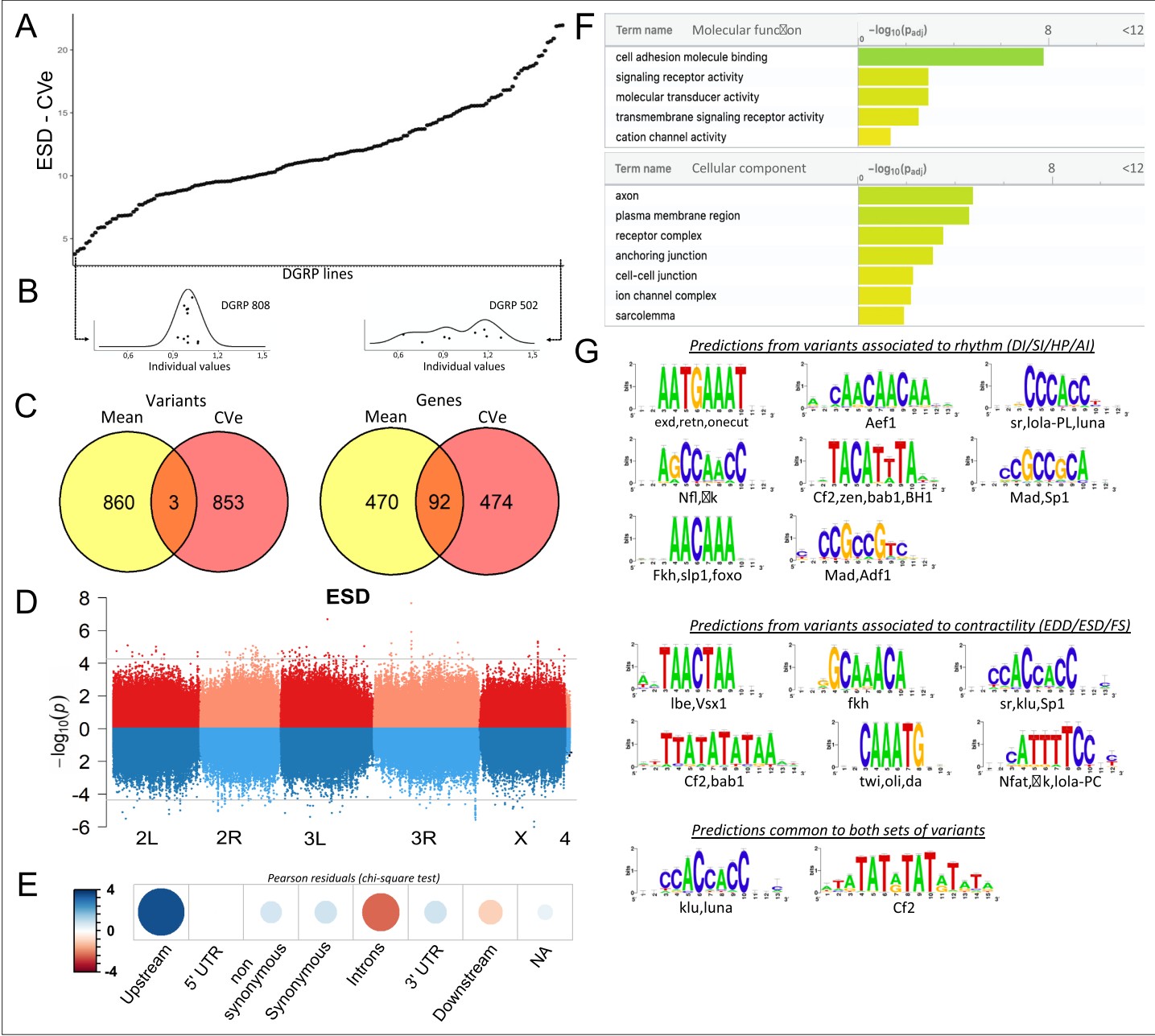

**Figure 4.** Quantitative genetics and genome-wide associations studies (GWAS) for cardiac traits CVe in the *Drosophila* Genetic Reference Panel (DGRP). (**A**) Distribution of end systolic diameter (ESD) CVe values among DGRP lines. Lines are arranged by increasing CVe values (other phenotypes are displayed in *Figure 4—figure supplement 1*). (**B**) Distribution of ESD values across individuals for two representative DGRP lines with low and high intra-genotypic variability. Each dot represents ESD value of a single fly. (Individual values scaled by line mean). (**C**) Venn diagram illustrating the overlap between single nucleotide polymorphisms (SNPs) (left) and genes (right) associated with mean (yellow) and CVe traits (pink). While only three SNPs were retrieved from GWAS of both mean and CVe traits, genes associated with CVe traits are largely overlapping with those associated with mean traits. (**D**) Miami plots showing the results of GWAS performed on mean of ESD (top, red) and ESD CVe (bottom, blue). No overlap between sets of variants is observed. Miami plots for other traits are displayed in *Figure 4—figure supplement 2*. (**E**) Pearson residuals of chi-square test from the comparison of indicated SNPs categories in the DGRP and among variants associated with cardiac traits CVe (see legend to *Figure 1C* for detailed description of SNPs categories). (**F**) Gene Ontology (GO) enrichment analyses of genes associated with CVe traits. Selected molecular functions and cellular components associated with variance of cardiac performances at FDR < 0.05 are shown. (**G**) Logo representation of position-specific scoring matrices (PSSMs) identified in corresponding sets of non-coding variants associated with cardiac traits CVe. Potential cognate transcription factors (TFs) indicated below were identified by comparing PSSMs to TF binding sites databases.

The online version of this article includes the following source data and figure supplement(s) for figure 4:

*Figure 4 continued on next page*

*Figure 4 continued*
**Source data 1.** Variants identified associated to phenotypic variance (Cve).
**Figure supplement 1.** Quantitative genetics of cardiac traits CVe.
**Figure supplement 2.** Comparison of genome-wide associations studies (GWAS) results for mean and CVe cardiac traits.

CVe of FS (*Figure 4—figure supplement 2*, *Supplementary file 3c*). The other variant, highlighted by both GWAS, was situated at more than 25 kb away from any gene. Despite unraveling largely distinct set of variants, GWAS analyses on CVe and trait means are associated with two sets of genes that overlap significantly (*Figure 4C*, *Supplementary file 3b*). Among 566 detected respectively from trait CVe and 562 genes detected by trait mean GWAS, 92 genes are found associated with both (FC=5, hypergeometric pval=$2.2 \times 10^{-39}$). Therefore, a significant number of genes contribute to natural variation of both mean and CVe traits, although through distinct variants.

Within the CVe-associated genes we observed an over-representation of cell adhesion processes, transmembrane signaling, and ion channel activity (pval=$1.6 \times 10^{-8}$ (FC=7), $1.1 \times 10^{-3}$ (FC=2,2), and $4.5 \times 10^{-2}$ (FC=2,5) respectively). Enriched cellular component annotations included cell-cell junction (pval=$5.7 \times 10^{-3}$ (FC=3.2), receptor complexes (pval=$3.3 \times 10^{-4}$ (FC=4.6), sarcolemma (pval=$1.3 \times 10^{-2}$ (FC=25), as well as several annotations related to neuronal projections suggesting both cell autonomous and non-autonomous activity of related genes (*Figure 4F*, *Supplementary file 3d*). In addition, genes leading to stress-induced lethality upon heart-specific KD (*Neely et al., 2010*) were significantly under-represented in this gene set (FC=0,6; p=0,05; *Supplementary file 3e*). This may suggest that a significant proportion of CVe GWAS-associated genes affect cardiac trait variance non-cell autonomously. As for trait means, variants localized in the proximal region of the genes (0–1 kb upstream of TSS) are over-represented among SNPs associated with traits CVe (*Figure 4E*). This suggested that natural variations of cardiac performance variance are also mainly affecting cis-regulatory regions. We also identified over-represented PSSMs in sequences within 150 pb of non-coding CVe variants (*Figure 4G*). Some of these identified PSSMs were specifically enriched in the sequences around CVe variants. This is the case for instance for Ladybird-early, Lbe, the fly orthologue of the mammalian LBX TF. However, several identified CVe PSSMs are similar to those identified in the vicinity of variants associated with trait means. This is the case for motifs predicted to be bound by Luna and Lola – variants of which are associated with CVe traits (*Supplementary file 3a*) – and it is also the case for Klu and Mad. Therefore, similar to the situation described for trait means, natural variations of phenotypic plasticity of cardiac performance appears, for a large part, to be driven by modifications of gene regulatory networks.

**Table 2.** Conserved genes associated with natural variations of cardiac traits from flies to humans. Enrichment analyses for genes conserved in human and for genes whose human orthologue is associated with either coronary artery diseases (CAD) or cardiac disorders. Numbers in parenthesis indicate the total number of genes in each analyzed gene set (whole fly genome/mean and CVe GWAS-associated genes). First row (human orthologue): Number of fly genes that display a human orthologue according to DIOPT (high and moderate rank). Second and third rows: Number of genes whose human orthologue (high and moderate rank) has been associated with CAD or cardiac disorders by genome-wide associations studies (GWAS) in human populations (*Supplementary file 3f-i*). Fold change (FC): Ratio between expected (based on successes observed on fly genome) and observed number of successes in respective GWAS gene sets. pval: hypergeometric p-value.

| | Fly genome (17500) | GWAS mean (562) | FC | pval | GWAS Cve (566) | FC | pval |
|---|---|---|---|---|---|---|---|
| Human orthologue | 9463 | 410 | 1.4 | $4.3 \cdot 10^{-21}$ | 379 | 1.25 | $7 \cdot 10^{-11}$ |
| CAD | 321 | 19 | 1.35 | 0.04 | 16 | 1.25 | 0.07 |
| Cardiac disorders | 944 | 68 | 1.66 | $7 \cdot 10^{-6}$ | 64 | 1.7 | $7 \cdot 10^{-6}$ |

## Conserved features of natural variations of heart performance between flies and humans

Next, we compared our GWAS results for cardiac performance in flies with similar data in humans. A literature survey identified a comprehensive set of human genes associated with cardiac disorders and coronary artery diseases (CAD) in human populations (*Supplementary file 3f and h*). Using DIOPT (*Hu et al., 2011*), we identified human orthologues of the genes associated with cardiac trait means and CVe in flies (*Supplementary file 4a and d*). Compared to the whole fly genome gene set (17,500 genes), both sets of GWAS-associated genes were significantly enriched for genes that have a human orthologue (FC = 1.4, pval = $4.3 \cdot 10^{-21}$ for genes associated with trait means and FC = 1.25; pval = $7 \cdot 10^{-11}$ for genes associated with traits CVe, *Table 2*), indicating that, overall, the loci associated with natural variations in cardiac traits have evolutionary conserved functions. More importantly, GWAS loci were enriched for genes whose human orthologues were associated with cardiac disorders (FC = 1.66 [trait means] and 1.7 [traits CVe], pval = $7 \cdot 10^{-6}$, *Table 2* and *Supplementary file 4b and e*). Lower enrichment was found for orthologues of human genes associated with CAD (FC = 1.35, pval = 0.04 [trait means] and 1.25, pval = 0.07 [traits CVe]). This analysis strongly supports the view that the cellular and molecular processes of heart development and physiology – which is already well characterized between fly and humans (*Ocorr et al., 2007b*; *Diop and Bodmer, 2015*; *Rosenthal et al., 2010*; *Bier and Bodmer, 2004*) – are not only conserved but also have a common genetic basis underlying their natural variation.

Building on these observations, we next wanted to determine whether our data in flies could guide the identification of novel players in the human cardiac system. We focused on the paired box TF gene *pox-meso* (*poxm*/PAX9) and on the zinc finger containing TF *stripe* (*sr*/EGR2) – both were associated with natural variations of cardiac trait means (*Supplementary file 1f*). Notably, neither the fly genes nor their human orthologues have a known function in the cardiac system. In flies, *poxm* contributes to the development and specification of somatic muscles (*Duan et al., 2007*), while *sr* is required for induction of tendon cell fate (*Gunage et al., 2017*). We first tested the effects of heart-specific KD on cardiac performance in flies using the in vivo assay. Both induced an increased heart rate (reduced HP and SI; *Figure 5A*) but did not affect other cardiac traits analyzed (not shown). The function of their human orthologues PAX9 and EGR2 was then tested in hiPSC-CMs. We asked whether siRNA-mediated KD affected action potential kinetics using a recently developed in vitro assay (*Figure 5B*; *McKeithan et al., 2017*). As shown in *Figure 5C–E*, both gene KDs induced shortening of the AP duration, thus suggesting conserved function in setting cardiac rhythm for both TFs. The effect was stronger for PAX9 KD and seems specific to the repolarization phase, as suggested by APD50 and 75 shortening (*Figure 5F–I*). APD shortening for PAX9 KD was coincident with increased expression of Na+ and K+ ion channels (*SCN5A, KCNH2,* and *KNCQ1*) (*Figure 5J*), supporting the APD shortening phenotype. In this context, the AP kinetics also correlated with shorter calcium transient duration (*Figure 5—figure supplement 1A-D and H-K*), including faster upstroke and downstroke calcium kinetics and increased beat rate (peak frequency) (*Figure 5—figure supplement 1E-G and L, M*), consistent with decreased expression of Calsequestrin 2 isoform (*CASQ2*) associated with PAX9 KD (*Figure 5J*). Finally, assessment of the PAX9 KD effect on sarcomeric content revealed an increase in sarcomeric gene expression (*Figure 5K*), and an upregulation of genes associated with an hypertrophic response (*NPPA, NPPB,* and *NPR1)* (*Battistoni et al., 2012*), which was coincident with increased CM size as quantified by the area of TNNT2 staining per cardiac nuclei (*Figure 5L and M*).

Collectively, these data illustrate conserved functions for *poxm*/*PAX9* and *sr*/*EGR2* in setting the cardiac rhythm and identify PAX9 as a novel and key regulator of cardiac performance at the cellular level, via the integrated regulation of expression of genes controlling electrophysiology, calcium handling, and sarcomeric functions in hiPSC-CMs.

## Discussion

Disentangling the relative contribution of genetics and environmental factors to cardiac pathologies and the influence of comorbidities is of fundamental importance from a medical perspective. From a fundamental perspective, natural selection is thought to operate on complex traits rather than on single genes, further illustrating the need to characterizing and understanding the genetic architecture of natural variations in cardiac performance. Although mammalian studies provide important

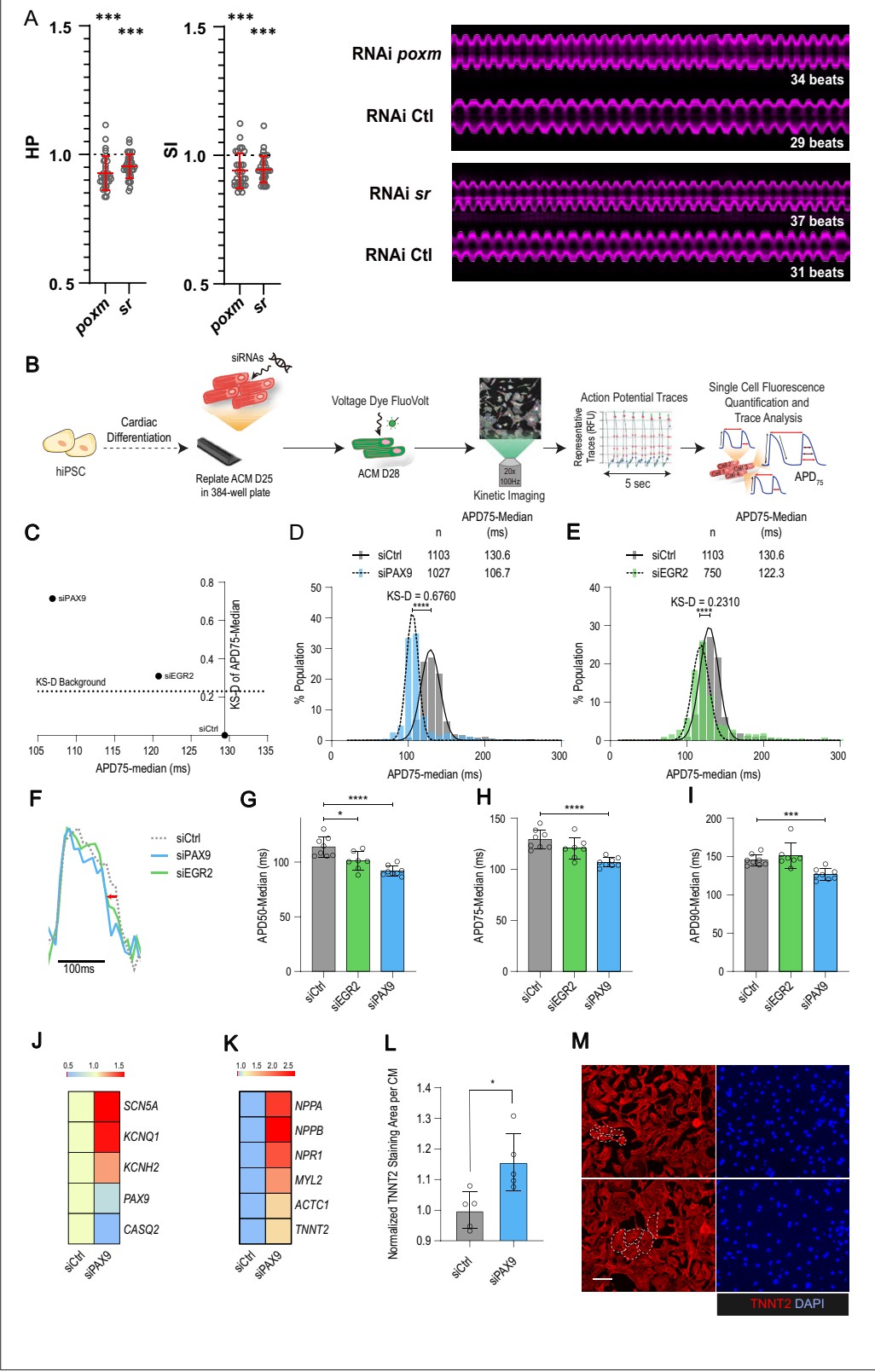

**Figure 5.** In vivo and in vitro assays for *poxm*/Pax9 and *sr*/Egr2 in flies and human induced pluripotent stem cells derived cardiomyocyte (hiPSC-CM). (**A**) Heart-specific knockdown (KD) of *poxm/Pax9* and *sr/Egr2* lead to increased heart rate in flies. Cardiac-specific RNAi KD phenotypes of tested transcription factors (TFs). Hand>Gal4; UAS RNAi 1-week adult females (n>30 per cross) were analyzed for cardiac performance using the in vivo assay on

*Figure 5 continued on next page*

*Figure 5 continued*

intact flies. Scatter plots of individual data, mean, and SD are represented. Wilcoxon test results are indicated (***: pval <0.005). Data were normalized to genotype matched controls. Right panel: representative M-modes (5 s) of RNAi KD and their respective control illustrating the increased heart rate observed upon *poxm* and *sr* inactivation. Number of heartbeats counted from M-modes are indicated. (**B–I**) Effects of Pax9 and Egr2 siRNA on hiPSC-CM action potentials. (**B**) Schematic overview of single cell and high-throughput voltage assay. (**C**) Two-dimensional graph for APD75 and Kolgomorov-Smirnov distance (KS-D) representing screen results for PAX9 and EGR2 KD. (**D–E**) Population distribution of APD75 measurements for siPAX9 and siEGR2 vs. siControl-transfected hiPSC-CMs, respectively. (**F**) Median action potential traces for siPAX9, siEGR2, and siControl-transfected hiPSC-CMs. (**G–I**) Histograms showing median APD50, APD75, and APD90 for siPAX9, siEGR2, and siControl-transfected hiPSC-CMs. n > 4 in all experiments. SD are represented. (**J,K**) Heatmaps of differentially expressed genes in siControl- and siPAX9-transfected hiPSC-CMs. (**L**) Histogram showing the effect of PAX9 KD on the average TNNT2 staining area per CM as compared to siControl. n > 4 in all experiments. SD are represented. (**M**) Representative images for siPAX9 and siControl. TNNT2 is shown in red and nuclei are stained with DAPI in blue. Scale bar = 25 µm. t-Test was used to calculate p-values. *p<0.05, ***p<0.001, ****p<0.0001.

The online version of this article includes the following figure supplement(s) for figure 5:

**Figure supplement 1.** Calcium transient measured in iPSC-ACM upon Pax9 and EGR2 siRNA-mediated knockdown (KD).

information about the molecular and cellular basis of heart development and physiology, these systems are less suited to dissecting the relationship between naturally occurring genetic variations and variations in individual cardiac performance.

Here, we report on the quantitative genetics of cardiac performance in *Drosophila*. We performed a standard GWAS of variants associated with cardiac trait looking at both within-line and intra-line variance for seven weakly correlated traits that also included an epistasis analysis. The cornerstone of our approach is a series of downstream analyses, which strongly implied that top hits are indeed genetic polymorphisms influencing quantitative variation in cardiac performance. These findings include: (i) the enrichment of pathways known to be related to cardiac development and physiology; (ii) the generation of an extensive network of protein-protein/genetic interactions that plausibly describes the architecture of cardiac performance; (iii) identification of significant overlap between gene occurrences for trait mean and variance; (iv) enrichment of likely causal variants near binding motifs for a set of cardiac-related TFs; (v) observation that several of these TFs are themselves GWAS candidate genes; and (vi) enrichment of fly genes for overlap with known human cardiac-related genes. In addition, we performed validation experiments that showed enrichment of cardiac effects for GWAS identified genes compared to randomly selected control genes. Importantly, we confirmed synergistic genetic interactions between the BMP and activin pathways; and defined two novel regulators of cardiac function that may well be relevant to human disease.

Our results therefore validate *Drosophila* as a unique model that permits the analysis of the etiology of the genetic foundations of cardiac function. Importantly, we identified many genes not previously associated with cardiac traits at the genetic, cellular, or molecular levels. This suggested that the space for natural variation associated with cardiac performance extends far beyond the genes and gene networks already identified as participating in the establishment and maintenance of cardiac function and illustrates the value of complementing classical genetic studies with an in-depth analysis of the genetic architecture of corresponding complex traits. A large number of candidate genes were validated in vivo using mutations or RNAi KD generated in unrelated genetic backgrounds, suggesting that the underlying biological processes are common to most genotypes.

Most of the variants identified affected non-coding regions of the genome. Recent studies in humans have highlighted that non-coding SNPs may affect the function of enhancers or promoters, thereby influencing the expression levels of surrounding or distant genes and which would explain how some non-coding SNPs may contribute to pathogenesis (*Tak and Farnham, 2015*; *Kikuchi et al., 2019*). Although we failed to find enriched potential TFBS at SNPs positions, extending the search space to sequences of ±75 bp around the variants revealed enriched DNA binding motifs, and pointed to potential roles for several TFBS. Our results indicate that variants likely affect sequences outside the core TF binding motif (e.g., flanking sequences) that in turn affect TF binding (*Grossman et al., 2017*). Supporting our findings, different reports on human GWAS analyses have shown that many risk-associated SNPs are not precisely located in conserved TF binding motifs but are in nearby

regions (*Heinz et al., 2013*; *Farh et al., 2015*). Several observations indicate that our predictions are accurate: (i) a large number of similar motifs were independently predicted from distinct sets of variants, (ii) some predicted motifs correspond to TFs whose role during cardiogenesis or for the maintenance of cardiac function have already been characterized, (iii) several discovered motifs correspond to TFs for which SNPs were themselves associated with the variation of some of the cardiac traits studied, and (iv) a high percentage of predicted motifs were validated by testing their cognate TFs in vivo for cardiac performance. Of note, while the cardiac GRN has been extensively studied in flies (*Seyres et al., 2016*; *Seyres et al., 2012*), a large proportion of the TFBS and cognate factors identified in this study have not been previously characterized as cardiac TFs, again illustrating the complementary contribution represented by the analysis of the genetic architecture of these phenotypes compared to more classical genetic, molecular, and cellular studies.

Although rarely studied, the genetic control of phenotypic variability is of primary importance. Indeed, variability offers adaptive solutions to environmental changes and may impact developmental robustness to micro-environmental perturbations (*Geiler-Samerotte et al., 2013*). From a medical genetics point of view, highly variable genotypes may produce individuals beyond a phenotypic threshold, leading to disease state (*Morgante et al., 2015*; *Gibson, 2009*). Having assessed the cardiac traits for multiple individuals from inbred lines, we were able to examine phenotypic variability among individuals with the same genotypes and therefore to analyze how natural variations impinge on such complex traits. Our results indeed highlight the important contribution that genetic control of variability can play in our understanding of the cause of phenotypic variation. An important feature concerns the nature of the variants and associated genes compared to trait means. In particular, genes identified by a global screen for genes cell autonomously affecting cardiac function (*Neely et al., 2010*) are under-represented among the genes associated with CVe traits. This suggests that this subset of genes influences the variance of cardiac traits in a cell non-autonomous manner. The separation between variants associated with trait means and those associated with the variance of the same traits was another remarkable feature of the genetic architecture of cardiac phenotypic plasticity. Although based on non-overlapping sets of non-coding regions, predictions of TFBS in the vicinity of non-coding variants identified overlapping sets of motifs. This suggests that natural variation of trait means and trait variance affect partially overlapping GRNs, albeit by distinct polymorphisms. Our analysis allowed us to identify common variants associated with phenotypic variability, and to identify essential characteristics of the genetic architecture of these traits. However, we cannot exclude that part of the observed variability is due also to rare alleles in partial LD with the lead associations (*Ek et al., 2018*) or to within-line accumulation of mutations. Nevertheless, the former is likely to be marginal, given the rapid decay of LD with local physical distance in *Drosophila* (*Mackay and Huang, 2018*).

Several previous studies have established that the principles identified on the DGRP population are universal and can be applied across phyla (*Anholt and Mackay, 2018*). In particular, the identification of human genes orthologous to those identified by GWAS in the DGRP may allow identification of a translational blueprint and prioritize candidate genes for focused studies in human populations (*Zhou et al., 2016*; *Carbone et al., 2016*). Our results provide additional support that the genetic architecture of quantitative traits is conserved. Indeed, a remarkable characteristic of the gene sets associated either with trait means or with trait variance is that they are both enriched for orthologues of human genes that are also associated with disorders affecting the cardiovascular system. This is a strong argument for considering that the genetic architecture of cardiac performance traits is conserved, at least in part, from flies to humans. Hence, the features learned from our study in flies should help reveal the characteristics of the natural variations of cardiac performance in human populations. One intriguing observation is that genes associated with intra-genotypic variance are highly significantly enriched for orthologues of human genes associated with cardiac disorders. Yet, GWAS on human populations are not designed to test the variants involved in intra-genotypic variations. One possible explanation may be that GWAS in humans can pick up variants that do not necessarily affect the mean of cardiac traits, but that cause extreme phenotypes in some individuals by affecting genes that function as buffers.

The high proportion of orthologous genes identified in GWAS in both flies and humans – for cardiac traits in *Drosophila* and for cardiac disorders in humans – suggested that our results in flies may accelerate discovery of causal genes in humans. We suggest that our study may be used as an aid in prioritizing genes at loci identified by GWAS for cardiac pathologies in humans. Indeed, we show

that our data allow identification of new actors that have a conserved role in cardiac function from flies to humans. Although not previously revealed by conventional approaches, we show that *poxm* and *sr* – two genes identified by GWAS as being associated with natural variations of cardiac traits – have a conserved function in the regulation of cardiac activity.

In summary, the analysis of natural variations of cardiac performance in *Drosophila*, combined with functional validations in vivo and in hiPSC-CM, represents a major advance in understanding the mechanisms underlying the genetic architecture of these complex traits, and holds promise for the identification of the genes and mechanisms underlying cardiac disorders in humans.

## Materials and methods

### Flies' strains and husbandry

Fly strains were obtained from the Bloomington *Drosophila* Stock Center or the Vienna *Drosophila* Resource Center. Flies were reared at a controlled density on corn flour/yeast/sucrose-agar medium at 25°C, 50% relative humidity and 12 hr light/dark. For movies acquisitions, newly emerged adults (0–24 hr) were collected and aged for 7 days. For screening of the DGRP population, 12 flies were analyzed for each DGRP strain. For validations (RNAi-mediated gene KD or mutant analyses), 30–40 flies per genotype were analyzed. All fly strains used in this study are listed in *Supplementary file 4h and i*.

### Fly heart-specific RNAi KD and mutant analyses

Driver-line (Hand>Gal4 4.2) (*Popichenko et al., 2007*) virgins were crossed to UAS-RNAi males or corresponding isogenic control males. BMP mutants were back-crossed into the same isogenic control background. Flies were raised on standard fly food and kept at 25°C. Female progeny were collected and aged to 1 week at 25°C, at which point they were imaged and analyzed.

### Semi-intact preparations and SOHA analyses

To measure cardiac function parameters, denervated, semi-intact *Drosophila* females adult fly hearts were prepared according to previously described protocols (*Ocorr et al., 2007c*). High-speed ~3000 frames movies were taken at a rate of 140 frames per second using a Hamamatsu ORCAFlash4.0 digital CMOS camera (Hamamatsu Photonics) on a Zeiss Axioplan microscope with a ×10 water immersion lens. The live images of the heart tube within abdominal A3-4 segments were captured using HCI imaging software (Hamamatsu Photonics). M-modes and cardiac parameters were generated using SOHA (http://www.sohasoftare.com/), as described previously (*Ocorr et al., 2007c*). M-mode records provide a snapshot of the movement of heart wall over time. Cardiac parameters were calculated as below: DI is the period when the heart is completely relaxed (diastole). SI is the period when the heart is actively contracted. HP is the time between the two consecutive diastoles (i.e., DI+SI) and heart rate is calculated from HP (1/HP). AI is the standard deviation of the HP mean for each fly normalized to the median HP. FS was calculated as (diastolic diameter – systolic diameter/diastolic diameter).

### In vivo analysis of heart function (tdtk)

R94C02::tdTomato (attP2) (*Klassen et al., 2017*) was combined with Hand-Gal4 as stable stock. Flies were then crossed to RNAi lines and reared to 3 weeks of age. To assay fluorescent hearts in vivo flies were mounted as described (*Klassen et al., 2017*) and were illuminated with green light (3 mW power). Five s movies of the beating heart were acquired at 270 fps and high-speed recordings were then processed using a custom R script (*Vogler, 2022a*).

### Source code for analyses

We have developed a GitHub project containing the instructions and material to reproduce the analyses (*Saha et al., 2021*). Source code are available in the GitHub repository. Required data and built singularity images are available on download. Instructions to reproduce the analyses are provided.

### Quantitative genetic analyses

Phenotype data were analyzed separately to perform quality control. In each line, data outside the range [Q1 − 1.5*IQR; Q3+1.5*IQR] were identified (where Q1/3 are first/third quartile and IQR = Q3

−Q1). Most of the films corresponding to these outlier data were re-analyzed. It was identified that these extreme results corresponded to technical problems (poor quality film, bad placement of the measurement points, etc.) and were removed from the analysis to avoid technical bias.

Mean and coefficient of variation (CV) were computed for each line and each phenotype. Only lines with at least seven remaining observations after QC were kept for the computation.

Correlation between phenotypes was computed using Spearman correlation between mean/CV of lines. Genetic indicators, like broad sense heritability, were computed from the QC data.

## Genotype-phenotype associations: GWAS and epistasis study

We used mean and CV summary statistics from our GWAS analysis to study the leverage effect of extreme values. We identified strong leverage effects in some phenotypes and treated them for outliers. We decided to remove lines with large leverage effect from GWAS analysis of the phenotype, since they imply a strong bias in the linear model. Hence, mean/CV values outside the range of [Q1 − 1.5*IQR; Q3+1.5*IQR] were discarded from GWAS analysis of the corresponding phenotype, thus removing all leverage effects.

GWAS analysis was performed using FastLMM (*Lippert et al., 2011*). Linear mixed model was built using Wolbachia infection and common polymorphic inversions (data provided by the DGRP Consortium) as covariates. A GitHub project containing the instructions and material to reproduce the analyses reported is available at: https://doi.org/10.5281/zenodo.5582846.

A QQ plot was generated to display the quantile distribution of observed marker-phenotype association p-values vs. the distribution of expected p-values by chance. The QQ plot was plotted to check if some polymorphisms are more associated with the phenotype under a null hypothesis, which would be represented by a uniform distribution of p-values. For each phenotype, we selected the 100 variants with the lowest GWAS p-value. These variants were mapped to genes using the following rules: (i) if the variant was located in a gene (INTRON, EXON, 3'or 5' UTR), the SNP was then associated with this gene, (ii) if not, the variant was associated with a gene if its distance from the boundary of the gene (TSS and TES was less than 1 kb); (iii) otherwise the variant was not mapped to a gene. Of the mapped variants we then examined variants with an MAF greater than 4% and discarded SNPs with synonymous effects. Epistasis analysis was performed using FastEpistasis (*Schüpbach et al., 2010*). For each phenotype, we used the SNP selected from the GWAS analysis as focal SNP and the interaction was tested with all the SNPs with MAF greater than 5% in the DGRP population. The best matching test SNP was selected for each focal SNP. Each resulting SNP was mapped to genes using the same rules as for GWAS variants.

Miami plots were designed according to *Vogler, 2022a*.

## Interaction network analysis

*Drosophila* interactome: A network generated from genetic and physical interaction databases has been built. Briefly, known protein-protein interactions, identified using Y2H and AP-MS techniques, and genetic interactions, are curated in the DroID database (http://www.droidb.org/) (*Murali et al., 2011*), to which the Y2H FlyBi dataset (https://flybi.hms.harvard.edu/results.php) has been added. Overall, this *Drosophila* interaction network contains 79,698 interactions between 11,022 genes/proteins. Network representations were made using Cytoscape (*Shannon et al., 2003*).

## Statistics

Enrichment p-values were based on a test following the hypergeometric distribution. Enrichment (FC) were calculated as the ratio between observed and expected successes in samples (based on the observations in the population). GO enrichment analyses were conducted using the online version of G:profiler tool (*Raudvere et al., 2019*) with default parameters, including g:SCS method for computing multiple testing corrections for p-values gained from GO and pathway enrichment analysis. (https://biit.cs.ut.ee/gprofiler/page/docs#significance_threshshold).

Statistical analysis of cardiac parameters in RNAi and mutant experiments were performed using R. The non-parametric Wilcoxon-Mann-Whitney test was used to compare the median of the difference between two groups (tested genes/respective control).

Comparing Z-scores (*Figure 2—figure supplement 1B*): For each phenotype and each gene, Z-scores summarizes the difference in the phenotype distribution from tested genes to their respective

control. Z-scores were calculated for all phenotypes (7 phenotypes) and all tested genes (GWAS-associated genes: 42, randomly selected genes: 18). The Z-scores for all the phenotypes of the tested genes were pooled together (GWAS-associated genes: 294 Z-scores; randomly selected genes: 126 Z-scores) and the distribution of absolute Z-scores between the two groups was analyzed using Mann-Whitney-Wilcoxon test.

Tests for genetic interactions (*Figure 2E* and *Figure 2—figure supplement 2*): Two-way ANOVA with interactions was conducted to test if the interaction of two genes is significantly different from the two-way ANOVA model with the additive effects of the two genes (*Kim, 2014*).

Phenotypic correlation between each trait pair was computed using Spearman correlation. Pearson chi-squared test was applied to test if the genomic location of variants associated with cardiac traits is biased toward any particular genomic region when compared against the whole set of variants in the entire genome of the DGRP population. The overlap coefficient/overlap index is a similarity measure that measures the overlap between two finite sets. This is used to quantify the overlap between the gene sets associated with the different cardiac traits.

## Motif's analysis

For every set of SNPs (rhythmicity CVe and mean, contractility CVe, and mean), the coordinates (dm3 genome assembly) were extended 75 nt to both sides and their corresponding fasta sequences were retrieved. TF binding motifs were identified using RSAT peak motifs (*Thomas-Chollier et al., 2012*) (default parameter) using two distinct background models: intrinsic (frequencies of short k-mers in the input sequences to calculate the enrichment of longer k-mers) and random (set of random genomic regions matching length and nucleotide composition using biasaway; *Worsley Hunt et al., 2014*). To reduce the redundancy of the found motifs (PSSMs), we applied the RSAT matrix-clustering (*Castro-Mondragon et al., 2017*) algorithm (default parameters) to find clusters of motifs. The clustered motifs were manually annotated by comparison with PSSM databases (Cis-BP *Weirauch et al., 2014*, JASPAR *Fornes et al., 2020*, and FlyFactorSurvey *Zhu et al., 2011*) using the tool SAT compare matrices (*Nguyen et al., 2018*) (default parameters).

## Generation of hiPSC-CMs

Id1 overexpressing hiPSCs (*Cunningham et al., 2017*) were dissociated with 0.5 mM EDTA (Thermo Fisher Scientific) in PBS without $CaCl_2$ and $MgCl_2$ (Corning) for 7 min at room temperature. hiPSC were resuspended in mTeSR-1 media (STEMCELL Technologies) supplemented with 2 µM Thiazovivin (STEMCELL Technologies) and plated in a Matrigel-coated 12-well plate at a density of $3\times10^5$ cells per well. After 24 hr after passage, cells were fed daily with mTeSR-1 media (without Thiazovivin) for 3–5 days until they reached ≥90% confluence to begin differentiation. hiPSC-ACMs were differentiated as previously described (*Burridge et al., 2015*). At day 0, cells were treated with 6 µM CHIR99021 (Selleck Chemicals) in S12 media (*Pei et al., 2017*) for 48 hr. At day 2, cells were treated with 2 µM Wnt-C59 (Selleck Chemicals), an inhibitor of WNT pathway, in S12 media. Forty-eight hr later (at day 4), S12 media is fully changed. At day 5, cells were dissociated with TrypLE Express (Gibco) for 2 min and blocked with RPMI (Gibco) supplemented with 10% FBS (Omega Scientific). Cells were resuspended in S12 media supplemented with 4 mg/L Recombinant Human Insulin (Gibco) (S12+media), 2 µM Thiazovivin and plated onto a Matrigel-coated 12-well plate at a density of $9\times10^5$ cells per well. S12+media was changed at day 8 and replaced at day 10 with RPMI (Gibco) media supplemented with 213 µg/µL L-ascorbic acid (Sigma), 500 mg/L BSA-FV (Gibco), 0.5 mM L-carnitine (Sigma), and 8 g/L AlbuMAX Lipid-Rich BSA (Gibco) (CM media). Typically, hiPSC-ACMs start to beat around day 10. At day 15, cells were purified with lactate media (RPMI without glucose, 213 µg/µL L-ascorbic acid, 500 mg/L BSA-FV, and 8 mM Sodium-DL-Lactate, Sigma), for 4 days. At day 19, media was replaced with CM media.

## Voltage assay in hiPSC-CMs

Voltage assay is performed using labeling protocol described in *McKeithan et al., 2017*. Briefly, hiPSC-CMs at day 25 of differentiation were dissociated with TrypLE Select ×10 for up to 10 min and action of TrypLE was neutralized with RPMI supplemented with 10% FBS. Cells were resuspended in RPMI with 2% KOSR (Gibco) and 2% B27 ×50 with vitamin A (Life Technologies) supplemented with 2 µM Thiazovivin and plated at a density of 6000 cells per well in a Matrigel-coated 384-well plate.

hiPSC-CMs were then transfected with siRNAs directed against PAX9 and EGR2 (ON-TARGETplus Human) using Lipofectamine RNAi Max (Thermo Fisher Scientific). Each siRNA was tested individually in 8-plicates. Three days post-transfection, cells were first washed with pre-warmed Tyrode's solution (Sigma) by removing 50 µL of media and adding 50 µL of Tyrode's solution five times using a 16-channel pipette. After the fifth wash, 50 µL of ×2 dye solution consisting in voltage-sensitive dye Vf2.1 Cl (Fluovolt, 1:2000, Thermo Fisher Scientific) diluted in Tyrode's solution supplemented with 1 µL of 10% Pluronic F127 (diluted in water, Thermo Fisher Scientific) and 20 µg/mL Hoescht 33258 (diluted in water, Thermo Fisher Scientific) was added to each well. The plate was placed back in the 37°C 5% $CO_2$ incubator for 45 min. After incubation time, cells were washed four times with fresh pre-warmed Tyrode's solution using the same method described above. hiPSC-CMs were then automatically imaged with ImageXpress Micro XLS microscope at an acquisition frequency of 100 Hz for a duration of 5 s with excitation wavelength of 485/20 nm and emission filter 525/30 nm. A single image of Hoescht was acquired before the time series. Fluorescence over time quantification and trace analysis were automatically quantified using custom software packages developed by Molecular Devices and Colas lab. Two independent experiments were performed.

## Calcium assay in hiPSC-CMs

Calcium assay is performed using labeling protocol as described (*Cerignoli et al., 2012*). Cells were prepared and transfected with siRNAs against PAX9 and EGR2 as described above in the voltage assay. Three days post-transfection, 50 µL of media was removed and replaced in each well by a ×2 calcium dye solution consisting in Fluo-4 NW dye (Invitrogen), 1.25 mM Probenecid F-127 (Invitrogen), and 100 µg/mL Hoescht 33258 (diluted in water, Thermo Fisher Scientific) diluted in warm Tyrode's solution (Sigma). The plate was placed back in the 37°C 5% $CO_2$ incubator for 45 min. After incubation time, cells were washed four times with fresh pre-warmed Tyrode's solution and imaged as described above.

## RNA-seq and data analysis

Day 25 hiPSC-CMs were transfected with 25 nM final concentrations of siRNA against PAX9 and with scrambled control siRNAs. After 3 days, cells were pelleted and resuspended in 500 µL TRIzol reagent followed by total RNA extraction. Library preparation and sequencing of the samples was done at La Jolla Institute of Immunology (La Jolla, CA). FASTQ files were processed using nf-core/rnaseq (*Ewels et al., 2021*) with Nextflow Version: 21.03.0.edge. Differential gene expression was determined using R/DESeq2 (*Love et al., 2014*) and GO term enrichment was done using gprofiler2 (*Kolberg and Raudvere, 2021*). All analysis scripts can be downloaded at https://github.com/gvogler/eLife-2022-Saha-et-al (copy archived at swh:1:rev:b102c48b85976216c37d2e5aed060670535192f6; *Vogler, 2022b*). All sequencing data were deposited on the Gene Expression Omnibus (GEO) repository (accession number GSE217655).

## Immunostaining of hiPSC-CMs

Three days post-transfection cells were fixed with 4% paraformaldehyde for 30 min and blocked in blocking buffer (10% Horse Serum, 10% Gelatin, and 0.5% Triton X-100) for 20 min. hPSC-CMs were stained with sarcomeric protein TNNT2 (Catalog #: HPA017888 Lot #: A91786, Sigma, dilution 1/200), secondary antibody Goat anti-Rabbit IgG Alexa Fluor 568 (Invitrogen, 1/500) and DAPI (1/1000) in Blocking Buffer. Cells were imaged with ImageXpress Micro XLS microscope (Molecular Devices) and custom algorithms were used to quantify the area of TNNT2 mask per cardiac nuclei. All experiments were performed at least in biological quadruplicates.

## Acknowledgements

The project leading to this publication has received funding from Fondation de France (N°00071034) and from Excellence Initiative of Aix-Marseille University - A*MIDEX (A*Midex International) to LP, from the National Institute of Health (R01 HL148827-03, R01 HL149992-02, R01 AG071464-01) and from the Department of Defence (W81XWH-21-1-0104-01) to AC and from National Institute of Health (R01 HL132241 and R21 AG061598) to KO. We thank the Bloomington *Drosophila* Stock Center (BDSC) and the Vienna *Drosophila* Research Center (VDRC) for fly stocks. Centre de Calcul Intensif d'Aix-Marseille is acknowledged for granting access to its high-performance computing resources.

## Additional information

### Funding

| Funder | Grant reference number | Author |
|---|---|---|
| Fondation de France | 00071034 | Laurent Perrin |
| Aix-Marseille Université | A*MIDEX | Laurent Perrin |
| National Institutes of Health | R01 HL148827-03 | Alexandre R Colas |
| National Institutes of Health | R01 HL149992-02 | Alexandre R Colas |
| National Institutes of Health | R01 AG071464-01 | Alexandre R Colas |
| U.S. Department of Defense | R01 AG071464-01 | Alexandre R Colas |
| National Institutes of Health | R01 HL132241 | Karen Ocorr |
| National Institutes of Health | R21 AG061598 | Karen Ocorr |

The funders had no role in study design, data collection and interpretation, or the decision to submit the work for publication.

### Author contributions

Saswati Saha, Data curation, Formal analysis, Visualization, Methodology, Writing - original draft, Writing - review and editing; Lionel Spinelli, Conceptualization, Formal analysis, Investigation, Visualization, Writing - review and editing; Jaime A Castro Mondragon, Formal analysis, Investigation, Methodology; Anaïs Kervadec, Michaela Lynott, Laurent Kremmer, Laurence Roder, Sallouha Krifa, Magali Torres, Investigation; Christine Brun, Resources, Data curation, Methodology; Georg Vogler, Resources, Software; Rolf Bodmer, Conceptualization, Funding acquisition; Alexandre R Colas, Karen Ocorr, Conceptualization, Supervision, Funding acquisition, Writing - review and editing; Laurent Perrin, Conceptualization, Formal analysis, Supervision, Funding acquisition, Investigation, Methodology, Writing - original draft, Project administration, Writing - review and editing

### Author ORCIDs

Lionel Spinelli  http://orcid.org/0000-0001-9228-8141
Georg Vogler  http://orcid.org/0000-0002-8303-3531
Rolf Bodmer  http://orcid.org/0000-0001-9087-1210
Alexandre R Colas  http://orcid.org/0000-0001-8489-0570
Karen Ocorr  http://orcid.org/0000-0003-2593-0119
Laurent Perrin  http://orcid.org/0000-0002-4139-8743

### Decision letter and Author response

Decision letter https://doi.org/10.7554/eLife.82459.sa1
Author response https://doi.org/10.7554/eLife.82459.sa2

## Additional files

### Supplementary files

• Supplementary file 1. Data on single marker GWAS performed on trait means.
(a) Tests for block effect within the 14 *Drosophila* Genetic Reference Panel (DGRP) lines analyzed twice. (b) 'Variants_MAF4%_merged': Summary table of variants with minor allele frequency (MAF) >4% identified by single marker GWAS for the seven traits, and their associated genes. (c) Summary table of genes associated with cardiac traits. (d) Analysis of genes found in at least two independent GWA analyses. (e) Summary table of genes associated with single nucleotide polymorphisms (SNPs) identified by FastEpistasis for traits means. Fly Base gene numbers (FBgn), gene symbol, and corresponding quantitative traits are indicated. Gene Ontology (molecular

function) annotations are provided. The 31 genes also found in single marker GWA (c) are labeled in red. (f) Summary of all trait mean GWA analyses associated genes. Summary table of genes associated with cardiac traits means, computing data from Fast LMM (single marker GWA), and FastEpistasis (Epistasis GWA). Fly Base gene numbers (FBgn), gene symbol, and method from which the associated gene was identified are indicated. Corresponding quantitative traits and Gene Ontology (molecular function) annotations are provided. The 31 genes found in both single marker and epistasis GWA are labeled in red.

• Supplementary file 2. Enrichments and annotations analyses of gene lists associated to trait means.
(a) Enriched annotations within GWA-associated genes traits means. gProfiler_allGWAS: Functional enrichment analysis performed on the 562 genes from FastLMM and FastEpistasis analyses, all traits merged (see *Supplementary file 1f*), using g:GOSt from G profiler (https://biit.cs.ut.ee/gprofiler/gost). Parameters used were as follows: Background: all annotated genes/selected GO terms size >5 and <500. (b) Enrichment analysis (hypergeometric distribution calculations) for genes identified in a global RNAi screen for heart function (*Neely et al., 2010*). (c) Principal network component (PNC) formed by genes and gene products from cardiac traits means GWAS (refer to *Figure 2B*). Details of the 148 genes forming the principal network components of direct genetic and physical interactions among the 562 cardiac performances-associated genes. Gene symbol, FBgn, and cardiac trait(s) for which they have been associated are indicated, together with corresponding Gene Ontology (GO) annotations. Annotations for signaling pathways (in biological processes) are highlighted in yellow, annotations for transcription factors (in molecular function) are highlighted in blue. (d) Details on the interactions depicted in *Figure 2B*. (e) Statistics of transcription factors (TFs) RNAi cardiac knockdown effects on heart performance traits. TFs were inactivated in the heart by Hand-Gal4 and heart parameters monitored in vivo by tdtk tomato high-speed video recording (see Materials and methods). Graphic representation of the data is displayed in *Figure 3C*. Wilcoxon-Mann-Whitney rank sum test results for indicated cardiac parameters are shown.

• Supplementary file 3. Data on GWAS for CVe traits and data from GWAS for human cardiac traits used for enrichment analyses.
(a) Summary table of genes associated with cardiac traits CVe, computing data from Fast LMM (single marker GWA), and FastEpistasis (Epistasis GWA). Raw data are provided in *Figure 4—source data 1*. Fly Base gene numbers (FBgn), gene symbol, and method from which the associated gene was identified are indicated. Corresponding quantitative traits and Gene Ontology (molecular function) annotations are provided. The genes found in both single marker and epistasis GWA are labeled in red. (b) List of the 92 genes found associated with both mean and CVe of cardiac performance traits. (c) Lists of all variants associated with CVe and mean identified by FastLMM and FastEpistasis and overlap between both lists. (d) Functional enrichment analysis performed on the 566 genes from FastLMM and FastEpistasis analyses, all CVe traits merged (a), using g:GOSt from G profiler. Parameters used were as follows: Background: all annotated genes/selected GO terms size >5 and <500. (e) Enrichment analysis (hypergeometric distribution calculations) for genes identified in a global RNAi screen for heart function (*Neely et al., 2010*). (f) Human genes found associated with coronary artery diseases (CAD) in genome-wide associations studies (GWAS). Human gene ID, SNP ID (when available), and original article describing the association are indicated. (g) *Drosophila* orthologues of human CAD genes identified using DIOPT. (h) Human genes found associated with cardiac traits in GWAS. Human gene ID, SNP ID (when available), quantitative trait analyzed for association, and original article describing the association are indicated. (i) *Drosophila* orthologues of human cardiac traits genes identified using DIOPT.

• Supplementary file 4. Comparisons and enrichment analyses between fly and human GWAS for cardiac traits.
(a) Fly cardiac mean genome-wide associations studies (GWAS) genes (*Supplementary file 1f*) and corresponding human orthologues identified with DIOPT. (b) Fly cardiac mean GWAS genes and corresponding human orthologues associated with cardiac disorders. (c) Fly cardiac mean GWAS genes, and corresponding human orthologues associated with CAD. (d) Fly cardiac CVe GWAS (*Supplementary file 3a*) genes and corresponding human orthologues identified with DIOPT. (e) Fly cardiac CVe GWAS genes and corresponding human orthologues associated with cardiac disorders. (f) Fly cardiac Cve GWAS genes and corresponding human orthologues associated with CAD. (g) Headcounts of fly mean GWAS genes for which human orthologues have been identified in human GWAS studies (moderate and high rank), and hypergeometric distribution calculations. (h) DGRP lines analyzed. (i) Other fly lines used in the study.

• MDAR checklist

## Data availability

All data generated or analysed during this study are included in the manuscript and supporting file and available on Zenodo (*Saha et al., 2021*).

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
