## [Editor Report]

The authors investigated natural variation and new genetic mechanisms underlying cardiac performance using sequenced inbred lines of the *Drosophila* Genetic Reference Panel. The study provides insights into the genetic architecture of complex cardiac performance traits and represents an important resource for researchers studying cardiac performance.

---

## [Decision Letter]

[Editors' note: this paper was reviewed by Review Commons.]

---

## [Author Response]

Overall, we acknowledge referee’s careful reading of the paper and comments that we think have helped further improvement of the manuscript.

On the attached pages are our detailed point by point responses to the referees’ comments along with a description of how the manuscript was modified in accordance.

New data included:

In response to the comments and suggestions of both reviewers 1 and 3, we conducted new experiments to test genetic interactions between different actors of the BMP and activin pathways. These new results confirm and complement the analyses described in the original manuscript. Furthermore, as suggested by reviewer 2, we have further studied the phenotypes of hiPSC-CM, by analyzing gene expression profiles and by analyzing the morphological changes induced as a result of PAX9 knockdown.

NB: The title has been slightly modified, to highlight the conserved features of the genetic architecture of cardiac performance revealed in the study

Former title: Genetic architecture of natural variation of cardiac performance in flies.

Novel title: Genetic architecture of natural variation of cardiac performance: From flies to humans.

Reviewer 11. The authors utilized the RNAi-mediated knockdown approach in their functional validation studies. It is not clear how each genetic variation (SNP) affects its associated genes. Could some of the SNPs activate the candidate gene expression? For the 4 candidate genes that failed to show cardiac defects, could the overexpression of these 4 genes alter cardiac performance?

Of course, we cannot predict direction of the effect of the variants on the function of the genes. In this context, loss-of-function experiments are subjected to a risk of false negatives. It is indeed possible that in the case of a lack of effect of the loss of function, a gain of function could reveal an effect. But gain-of-function experiments are difficult to control, and often subjected to non-specific effects because it is complicated to control the level of over-expression compared to endogenous expression. This did not seem suitable for an extensive analysis of a large number of genes. We therefore chose to test only for loss of function.

In addition, our approach to testing heart-specific RNAi aims to assess the quality of the association results by comparing RNAi for genes identified by GWAS to randomly selected genes. It is not intended to describe precisely the involvement of each gene individually.

(See also answer to reviewer 2 comment n°2 and the modifications to the manuscript that have been made and which address these criticism).

2. babo is the type I activin receptor, not type 2.

Thank you, we have corrected this error.

3. The authors show BMP and activin pathway genetically interacts to affect cardiac performance. But it is interesting to find that these interactions are in a trait-dependent manner. For example, it seems that babo and dpp epistatically interact to regulate FS, while they additively regulate HP and DI. The authors need to discuss the complex genetic interaction further.

See reply to reviewer 3, comment N°2 below.

4. Both snoo and sog are identified from GWAS. How about babo and dpp? Are there any identified SNPs associated with babo and dpp?

Considering GWAS for mean phenotypes, there is no variant in *dpp* that are within the 100 best ranked SNPs nor within the variants identified using fast epistasis. But given the size of the DGRP population we are far from being exhaustive, as we do not reach saturation. It is therefore difficult to comment on these ‘negative’ results. However, we do identify one variant in *babo* using fast epistasis (see figure 2B and Table S3).

5. It is unclear why the mad KD behaves oppositely to dpp mutant, although both proteins are involved in BMP pathway. In Figure S5, the mad KD shows reduced FS and HP, but dpp LOF mutant shows increased FS and HP (Figure S4). Can the authors perform RNAi to knockdown dpp specific in the heart to reexamine the role of dpp in the regulation of cardiac function. The whole body LOF mutant dpp-d14 might not target cardiac tissue directly to control heart performance like mad KD.

(See also answer to reviewer 3 comment n°2) We did perform heart specific *dpp* RNAi experiments together with other tests for interactions using new allelic combinations of activin and BMP pathways and therefore can compare heart specific knock down to heterozygotes for amorphic mutations for both *dpp* and *mad*.

Regarding *dpp*, congruent effects on HP, DI, SI, ESD and EDD were observed between mutant and RNAi, while RNAi had opposite effects on FS compared to heterozygotes *dppd^14^* mutants (decreased and increased FS compared to control, respectively). In the case of *mad*, heterozygous mutants had no effect on FS, EDD and ESD, but similarly to *dpp* mutants it increased SI, DI and HP. *mad* RNAi uniquely decreased HP, DI and SI and increased AI. However, similarly to *dpp* RNAi, it induced a decrease of FS.

Thus, systemic versus heart specific knockdown of genes induce specific effects, suggesting cardiac non-autonomous interactions. This complex picture of TGFb involvement is now discussed in the result section (see below, Reviewer 3, major comment 2).

6. The authors selected two novel genes to study the conversed regulation in both flies and human iPSC cells. Besides testing these novel genes, the authors should also verify whether the conserved pathways, like TGF-β, regulate heart performance in human iPSC cells similar to the flies.

We focused on poxm/Pax9 and sr/Egr2 because none of these TFs were known to have cardiac function in fly nor in mammals. Our paralleled analyses in fly and hiPS-CM illustrates how the description of the genetic architecture of cardiac traits in flies can accelerate discovery in mammals.

There is extensive literature describing the involvement of TGF B /BMP and Activin pathways in heart development and diseases in humans, hence the choice not to focus on these pathways in iPS-CM.

Reviewer 2:1**.** It will be interesting to compare this fly GWAS to human heart disease GWAS data (for example, cardiomyopathy, arrhythmia, heart failure) from patients. Such cross comparison could make the data set more valuable.

We actually did make this comparison (Table 2, Table S11) and we agree it significantly validates our approach. This identified a set of orthologous genes associated with cardiac traits both in *Drosophila* and humans, supporting the conservation of the genetic architecture of cardiac performance traits, from arthropods to mammals.

2. RNAi is the only experimental approach in this manuscript to validate the functional significance from data analyses. Authors may consider using genetic mutations such as deficiency lines or P-element lines to offer an alternative approach. This is simply a suggestion to improve the rigor and reproducibility, not absolutely required.

In an attempt to provide a consistent analysis of loss of gene function, our strategy was to concentrate our analysis on the effects of heart specific knock down. This allows us to compare -in a global way- the effects of the knock down of genes identified by GWAS to those of randomly selected genes.

Our objective was to provide a global view of the heart specific effects of the identified genes, and not to characterize precisely the involvement of each of them, using a combination of mutant alleles, RNAi and gain of function. Given the experimental burden of analyzing cardiac function, such a strategy would have indeed required us to concentrate only a very small number of genes.

We however recognize that this strategy has limitations:

– Some variants may lead to gain-of-function effects of genes, and our strategy is not able to test for these effects.

– Some variants may come from non-cell-autonomous effects, which would not be replicated by our targeted RNAi strategy in the heart.

Therefore, the false negative rate of our experiments is difficult to estimate.

We have tried to put this into perspective and to highlight the limitations of our analysis in the Results section describing RNAi validation of GWAS results.

“To assess in an extensive way whether mutations in genes harboring SNPs associated with variation in cardiac traits contributed to these phenotypes […] These results therefore supported our association results. It is important to emphasize that our approach is limited to testing the effect of tissue-specific gene knock down. Since some of the variants may lead to increased gene function and/or expression, this can lead to a false negative rate that is difficult to estimate. In addition, some of the associated variants may influence heart function by non cell-autonomous mechanisms, which would not be replicated by cardiac specific RNAi knock down.”

3**.** In order to validate the roles of predicted TF binding sites, the best approach would be introducing point mutations using CRISPR/Cas9 within the binding motif then testing out molecular and physiological outcomes. Rather authors chose to test indirectly to knock down those TFs. If so, authors need to at least acknowledge the potential caveats of such approach and the limitation in related data interpretation.

The reviewer is right, the definitive proof of the involvement of a potential TF binding site on the regulation of a gene located in cis requires to mutate the binding site and to analyze the effect on the expression of the corresponding gene. But this may not be sufficient to definitely demonstrate that the potential TF is indeed a regulator of that gene (the binding motif may be target of yet another TF): definitive proof may require motifs/TF DNA binding domain swaps. This would have been out of the scope of the present study. In addition, the effects on heart performance of mutating one TFBS at a time (among several dozens) may be too weak to allow their characterization with available tools and approaches.

We acknowledge however that our approach provides an indirect validation of transcription factors binding sites predictions. This was, in our opinion, the most efficient way to evaluate the potential effect of predicted transcription factors.

We clarify this in the result section:

“We did not test individually the effects on cardiac performance of mutations in predicted TFBSs located near the SNPs because any individual effect would probably be too small to be detectable by the available methods. Rather, we tested the potential involvement of their cognate TFs by cardiac specific RNAi mediated KD”

4**.** hiPSC-CM data is somewhat limited by only showing the HR and AP duration data. It is recommended to include some immunocytochemistry data to show the morphology, sarcomere structure of these hiPSC-CMs. Gene expression data generated by qPCR or RNA-seq in particular focusing CM structure and function genes would be helpful too.

As suggested by referee 2, we have now performed gene expression analysis and immunostaining of PAX9 KD which gave the strongest phenotype in iPSC-CM (Figure 4 J-M). This unraveled increased expression of Na^+^ and K^+^ channels, which is in line with APD shortening phenotype, as well as down regulation of CASQ2, consistent with calcium transient shortening. Expression analysis also revealed increased sarcomeric genes and NPPA/B expression, which was consistent with increased CM size as quantified by the area of TNNT2 staining per nuclei.

These new data are described at the end of the result section:

“APD shortening for PAX9 KD was coincident with increased expression of Na^+^ and K^+^ ion channels *(SCN5A, KCNH2* and *KNCQ1*) (Figure 4J), supporting the APD shortening phenotype. In this context, the AP kinetics also correlated with shorter calcium transient duration (Figure S8A-D and H-K), including faster upstroke and downstroke calcium kinetics and increased beat rate (peak frequency) (Figure S8E-G and L, M), consistent with decreased expression of Calsequestrin 2 isoform (*CASQ2)* associated with PAX9 KD (Figure 4J). Finally, assessment of the PAX9 KD effect on sarcomeric content revealed an increase in sarcomeric gene expression (Figure 4K), and an upregulation of genes associated with an hypertrophic response (*NPPA, NPP*B and *NPR1*) (Battistoni Et al. Circulating biomarkers with preventive, diagnostic and prognostic implications in cardiovascular diseases, *Int J Cardiol*, 2012, vol. 157) which was coincident with increased CM size as quantified by the area of TNNT2 staining per cardiac nuclei (Figure 4 L, M).

Collectively, these data illustrate conserved functions for *poxm*/*PAX9* and *sr*/*EGR2* in setting the cardiac rhythm and identify PAX9 as a novel and key regulator of cardiac performance at the cellular level, *via* the integrated regulation of expression of genes controlling electrophysiology, calcium handling and sarcomeric functions in hiPSC-CMs.”

Reviewer 3Major Comments:1. There is an assumption in the use of RNAi knockdown to validate the genes identified in the quantitative analysis, and that is that natural variants are themselves hypomorphic. It is possible that among the variants identified some are hypermorphic, or among the transcription factor binding sites that variants lead to increased factor binding. While RNAi knockdown is an excellent choice to begin validation, I do not think the authors can rule out that a gene not functionally validated by their RNAi tests does not have a role in cardiac function.

Please see our answers to reviewer 1 comment n°1 and reviewer 2 comment n°2.

2**.** After performing RNAi knockdown to validate genes identified by GWAS the authors focus on the TGFbeta signaling pathway for downstream analysis. To do so they examine heterozygotes for sog, a repressor of BMP signaling, and snoo, an activator of Activin pathway. The data from the snoo/sog heterozygote is compelling in its disruption of heart phenotypes, and the authors conclude a "coordinated action of activin and BMP." snoo, however, also works as a transcriptional repressor in the BMP pathway, so it's possible that the effects the authors are seeing here could be confined to an increase in BMP signaling. Unlike snoo and sog, mutations in babo and dpp are both expected to have negative effects on Activin and BMP signaling, respectively. The babo/dpp interaction is not as quantitatively convincing as the snoo/sog data, despite the integral roles both babo and dpp play in their respective pathways. If both pathways are connected, why do snoo/sog heterozygotes affect SI phenotypes, while babo/dpp heterozygotes affect fractional shortening? I think the authors data suggest an interesting potential interaction between these pathways, which could be confirmed by examining further mutant combinations, knockdowns or increased expression transgenes, but falls short of a "confirmed synergystic genetic interaction." It does, however, underscore the value of the data in the paper for opening up new avenues for future study.

(and reviewer 1 comments 3 and 5).

These comments led us to reconsider the analysis of the phenotypes associated with loss of function of the TGFb pathway, and to analyze other pathway components combinations.

We acknowledge reviewer 3 criticisms on *snoo/sog* experiments, which are difficult to interpret given the broad action *snoo* may have on both BMP and activin pathways. We have addressed this in the result section.

We have also analyzed other allelic combinations of BMP and activin pathways components, which strengthen the analysis performed on *dpp/babo*. Indeed, we tested *babo/tkv* heterozygotes (respectively specific activin and BMP receptors) and found significant genetic interactions for ESD and EDD. Albeit non-significant, *babo/tkv* double heterozygotes display a tendency to non-additive effects on FS (p= 0,054). *mad/smox* heterozygotes (respectively specific downstream TFs of BMP and activin pathways) display interactions (non-additive effects) on HP, SI, DI, ESD and EDD. These new results (Supplemental Figure 4) are thus supporting the hypothesis of genetic interactions between the pathways, but also reveal, as suggested by reviewer 3, a complex relationship between both pathways since interactions are revealed for specific traits in each of the mutant combinations analyzed.

The phenotypes related to the individual loss of function of each of the actors of these pathways (*dpp*, *tkv* and *mad* for BMP; *babo* and *smox* for activin) are however very similar. When they have an effect, heterozygous amorphic alleles of these genes display increased phenotypes related to rhythmicity (HP, DI, SI, AI) and FS, but decreased cardiac diameters (ESD and EDD).

Finally, as pointed out by reviewer 1, the picture is certainly even more complex since the phenotypes of RNAi mediated heart specific loss of function are not always similar to those of systemic loss of function. Indeed, *mad* RNAi causes a reduction of HP, DI, SI and FS (Figure S5) whereas heterozygotes for *mad^12^* have either no or opposite effect on these phenotypes, and *mad* RNAi causes a significative increase in AI whereas *mad^12^* has no effect (Figure S4). The discrepancy between tissue specific RNAi and heterozygous background was also found in the case of *dpp*, but specifically for the FS. Indeed, as suggested by reviewer 1 we have analyzed the loss of function of *dpp* by heart-specific RNAi. *dpp* RNAi results in a reduction of the FS (like *mad* RNAi) whereas the loss of function in the whole-body results in an increase of the FS.

We therefore re-wrote the whole corresponding section of the results and modified Figure S4 to include *babo/tkv; smox/mad* and *dpp* RNAi data.

“We further focused on the TGFb pathway, since members of both BMP and activin pathways were identified in our analyses. We tested different members of the TGFb pathway for cardiac phenotypes using cardiac specific RNAi knockdown (Figure 2C), and confirmed the involvement of the activin agonist *snoo* (Ski orthologue) and the BMP antagonist *sog* (chordin orthologue). Notably, Activin and BMP pathways are usually antagonistic (Figure 2D). Their joint identification in our GWAS suggest that they act in a coordinated fashion to regulate heart function. Alternatively, it may simply reflect their involvement in different aspects of cardiac development and/or functional maturation. In order to discriminate between these two hypotheses, we tested if different components of these pathways interacted genetically. Single heterozygotes for loss of function alleles show dosage-dependent effects of *snoo* and *sog* on several phenotypes, providing an independent confirmation of their involvement in several cardiac traits (Figure S4). Importantly, compared to each single heterozygotes, *snoo^BSC234^/ sog^U2^* double heterozygotes flies showed non additive SI phenotypes (two-way ANOVA p val: 2,1 10-7) suggesting a genetic interaction (Figure 2E and Figure S4A). It is worth noting however that *snoo* is also a transcriptional repressor of the BMP pathway (PMID: 16951053). The effect observed in *snoo^BSC234^/ sog^U2^* double heterozygotes can therefore alternatively arise as a consequence of an increased BMP signaling without affecting the activin pathway. We thus tested other allelic combinations for loss of function alleles of BMP and activin pathways. *babo/tkv* heterozygotes (respectively activin and BMP type 1 receptors) displayed non additive ESD and EDD phenotypes (Figure S4C). Synergistic interaction of BMP and activin pathways was also suggested by the analysis of fractional shortening in loss of function mutants for *babo* and *dpp*, the BMP ligand (Figure S4B). Of note, *babo/tkv* double heterozygotes also displayed a tendency to non-additive effects on FS albeit non-significant (two-way anova p= 0,054). In addition, *mad/smox* heterozygotes (specifc downstream TFs of BMP and activin pathways) displayed non-additive effects on several traits, including phenotypes related to rhythmicity (HP, SI, DI) and contractility (ESD and EDD) (Figure S4D). Altogether, cardiac performance in response to allelic combinations of activin and BMP supported a coordinated action of both pathways in the establishment and/or maintenance of cardiac activity. This was further supported by the observation that simple heterozygotes for the tested loss of function alleles displayed similar trends with respect to cardiac performance, irrespective of the pathway considered (*dpp, tkv* and *mad* for BMP; *babo* and *smox* for activin). Indeed, they displayed either no effect or increased fractional shortening and rhythmicity phenotypes (HP, DI, SI, AI), and decreased cardiac diameters (ESD and EDD). This suggests coordinated activity of both pathways. Importantly, the genetic interactions were tested using amorphic alleles that lead to systemic loss of function. The observed phenotypes may thus not unravel cardiac specific effects of the pathways. In support of this, *mad* cardiac specific RNAi knock down was tested (see below, Figure S5) and lead to a decreased HP, DI, SI and FS whereas heterozygotes for *mad^12^* have either no (FS) or opposite (HP, DI, SI) effect on these phenotypes (Figure S4D). Inversely, *mad* RNAi caused a significant increase in AI whereas *mad^12^* had no effect. However, heart specific *dpp* RNAi knock down (Figure S4E) lead to similar phenotypic trends compared to *dpp^d14^* (increased HP, DI, SI, decreased EDD and ESD) with the notable exception of FS which was reduced following cardiac specific KD (Figure S4E), but increased in *dpp^d14^* heterozygotes (Figure S4B). Taken together, these data point to a complex picture of TGFb pathway activity in regulating cardiac performance, involving both the activin and the BMP pathways as well as gene specific effects with both systemic and tissue-specific contributions.”

Minor Comments:There is an enormous amount of data in this paper, but there are places where things are summarized a little too briefly. For example, there are no definitions given at the beginning of the Results section for traits like "Heart Period" or "Systolic Interval," which would make this work significantly more accessible for other *Drosophila* researchers. (They do touch on this when they explain later in the paper that certain variants are "associated with quantitative traits linked to heart size and contractility" but more background earlier would be helpful.) When we consider heart performance traits, what is the baseline from known mutants? In other words, where is the line between variation and defect?

– We have detailed the description of the traits analyzed at the beginning of the result section. We hope this improves the ease of reading in the direction suggested by the reviewer.

“7 cardiac traits were analyzed across the whole population (Dataset S1 and Table 1). As illustrated in Figure 1A, we analyzed phenotypes related to the rhythmicity of cardiac function: the systolic interval (SI) is the time elapsed between the beginning and the end of one contraction, the diastolic interval (DI) is the time elapsed between two contractions and the heart period (HP) is the duration of a total cycle (contraction + relaxation (DI+SI)). The arrhythmia index (AI, std-dev(HP)/mean (HP)) is used to evaluate the variability of the cardiac rhythm. In addition, 3 traits related to contractility were measured. The diameters of the heart in diastole (End Diastolic Diameter, EDD), in systole (End Systolic Diameter, ESD), and the Fractional Shortening (FS), which measures the contraction efficacy (EDD-ESD/EDD).”

– With respect to the baseline of cardiac performance, there is no simple answer. The baseline is influenced by the genetic background and the experimental conditions. This is the reason why any analysis of mutants or RNAi is conducted in comparison with its own control, analyzed at the same time. Concerning the DGRP lines, no baseline can be defined, since the objective is to measure the diversity of cardiac performance traits within a natural population.